# From Signaling Pathways to Distinct Immune Responses: Key Factors for Establishing or Combating *Neospora caninum* Infection in Different Susceptible Hosts

**DOI:** 10.3390/pathogens9050384

**Published:** 2020-05-16

**Authors:** Ragab M. Fereig, Yoshifumi Nishikawa

**Affiliations:** 1National Research Center for Protozoan Diseases, Obihiro University of Agriculture and Veterinary Medicine, Inada-cho, Obihiro, Hokkaido 080-8555, Japan; ragabfereig2018@gmail.com; 2Department of Animal Medicine, Faculty of Veterinary Medicine, South Valley University, Qena City, Qena 83523, Egypt

**Keywords:** *Neospora caninum*, neosporosis, immunity, signaling pathways, infection

## Abstract

*Neospora caninum* is an intracellular protozoan parasite affecting numerous animal species. It induces significant economic losses because of abortion and neonatal abnormalities in cattle. In case of infection, the parasite secretes numerous arsenals to establish a successful infection in the host cell. In the same context but for a different purpose, the host resorts to different strategies to eliminate the invading parasite. During this battle, numerous key factors from both parasite and host sides are produced and interact for the maintaining and vanishing of the infection, respectively. Although several reviews have highlighted the role of different compartments of the immune system against *N. caninum* infection, each one of them has mostly targeted specific points related to the immune component and animal host. Thus, in the current review, we will focus on effector molecules derived from the host cell or the parasite using a comprehensive survey method from previous reports. According to our knowledge, this is the first review that highlights and discusses immune response at the host cell–parasite molecular interface against *N. caninum* infection in different susceptible hosts.

## 1. Introduction

Neosporosis is a clinical protozoan parasitic disease caused by *Neospora caninum*, the intracellular apicomplexan parasite. Such a parasite is similar to another apicomplexan parasite, *Toxoplasma gondii*, the well-studied protozoan parasite. The disease affects a large number of warm-blooded animals and is transmitted orally via the ingestion of oocysts or tissue cysts, or from an infected dam to the fetus by transplacental transmission. Such an infection is incriminated as a cause of abortion in cattle worldwide, leading to high financial burdens and losses in cattle industry [1]. Shortly after infection, it is perhaps tachyzoites that are the first parasitic stage which efficiently make contact with host immune effectors, seeking the successful establishment of the infection [2]. However, the outcomes of *N. caninum* infection are greatly variable depending on the type of host, mode of infection, physiological factors (age, sex, pregnancy), and the parasite. Even in the same host with similar physiological statuses, the sequelae of infection may be varied, supporting evidence of an essential role for the immune system [3]. Generally, the immediate immune response against *N. caninum* infection involves the activation of antigen presenting cells (APCs), especially macrophages and dendritic cells (DCs), aided with interferon-gamma (IFN-γ) which are incorporated in the production of high amounts of pro-inflammatory mediators. As a response to this inflammatory milieu, tachyzoites differentiate quickly to bradyzoites (dormant stage) which can hide from the host arsenals by representing in immune effectors [4]. Although there is no evidence for neosporosis in humans, the disease recently gained significant interest because of the massive economic losses associated with the abortion of cattle [1]. Immune response-related studies against the *N. caninum* infection, especially at the molecular level, have increased in the last decade and some reviews have also highlighted such information and findings. Aguado-Martinez et al. (2017) [5] reviewed the crosstalk between *N. caninum* infection and host–immune response, and provided useful information on the manipulation of innate and adaptive immune effector molecules against infection in pregnant and non-pregnant mice. However, this review focused mainly on such mechanisms in a mouse model. Other reviews have also collected data related to the immune response against *N. caninum* [6,7,8,9,10]. However, in the current review, we will discuss cellular and humoral immunities with a special focus on the recognition and signaling pathways of host cells and triggering parasite factors in a comprehensive survey method of previous reports. An extensive search was performed in the PubMed database for articles that included the search keywords “*Neospora caninum* infection” and “neosporosis” in their title and abstract to be recorded. Studies discussing all aspects of infection, pathogenesis, immunity, or host–*Neospora* interactions were thoroughly investigated. Then, only studies that focused on host–parasite interactions from various immunological approaches were included, with special reference to recognition and signaling pathways. According to our knowledge, this is the first review that highlights and discusses immune response at the host cell–parasite molecular interface against *N. caninum* infection in different susceptible hosts.

## 2. Overview of Recent Researches on the Machinery of Neosporosis

The advent of -omics data—such as genomics, transcriptomics, and proteomics—has led to potential advances in terms of understanding the host–pathogen interaction. These research fields have resulted in a great discovery of potential host effectors and signaling pathways concerned with the combating of infection. Similarly, various parasite derived-molecules were identified that can manipulate host factors and, amazingly, establish successful infections. In the case of *N. caninum*, these branches of research are in relatively preliminary steps because of the discovery of the parasite at end of 20th century (identified in 1988) [3]. In *T. gondii*, researchers had achieved great steps in aforementioned approaches and thus numerous clues of infection and host–parasite interactions have been solved. For this reason, *T. gondii* is regarded as a potential and fascinating model pathogen for studying the research concerned with pathogenesis, immune response, and aspects of host–parasite interactions for many intracellular protozoan parasites particularly *N. caninum* [5]. Therefore, herein we might exploit the wealth of information available for *T. gondii* in concluding those we obtained for *N. caninum* relating to aspects of parasite organelles functions and morphology, pathogenesis and host–parasite interactions as discussed in later sections.

## 3. Historical View and Evolution of *N. caninum*

In Norway in 1984, Bjerkas et al. (1984) [11] was first identified as a causative agent of parasite origin to paralytic cases in domestic dogs. Later, such a parasite was identified as a new species in 1988 and named as *N. caninum* [12]. In an earlier time (before 1988), *N. caninum* was misdiagnosed as *T. gondii* due to their high morphological and developmental similarities [13,14]. However, numerous differences have been reported between *T. gondii* and *N. caninum* in many aspects. *N. caninum* and *T. gondii* are coccidian parasites which belong to the phylum Apicomplexa and contain an additional three clearly defined groups: gregarines (e.g., *Lankesteria*); haemosporidians (e.g., *Plasmodium*); and piroplasms (e.g., *Theileria*, *Babesia*) [15]. Although there are substantial variations among such groups, the many similarities between apicomplexan species indicate that they all share a common ancestor. The apicoplast is an essential part of *N. caninum*; most apicomplexan parasites have an interesting biological history that provides an insight into the evolution of such parasites. This organelle evolved from an interaction between the parasite and red algae, either by engulfment or invasion [16,17]. Researchers suggest that *N. caninum* and *T. gondii* diverged from their common ancestor in a period between 12 and 80 million years ago based on ssu rRNA analysis. Perhaps this divergence was caused by the natural selection of definitive hosts. Further speciation of *N. caninum* most likely occurred somewhere between 12 and 57 million years ago [18].

## 4. Ultrastructure of *N. caninum* and Functions of Essential Organelles

There are three developmental stages of *N. caninum* capable of inducing infection: tachyzoite, bradyzoite, and sporozoite. Zoites or cell invasive stages have a crescent shape and are almost similar in basic structure. *N. caninum* is a single celled-organism, however, it possesses well-structured and accommodated organelles, rendering it as a proficient pathogenic parasite in a wide range of animal hosts. Apical secretory organelles such as rhoptries, micronemes, and dense granules are considered of special concern in *N. caninum* and *T. gondii* because of their professional role in the development, invasion, and survival of the parasite inside the host cell [19,20]. The protein secretions from these organelles assist in the invasive stages of the parasite to attach, invade, adapt, and establish infection within the host cell [21,22,23]. In addition, the initial host cell recognition of the parasite that is mediated by parasite surface antigens is a critical step for establishing successful infection [24].

### 4.1. Surface Antigens

*N. caninum* tachyzoites surface antigens regulate the process of adhesion and invasion of host cells. Additionally, they contribute to the interaction of the parasite with the immune system, and subsequently the pathogenesis of the disease. Two major surface antigens were identified and widely studied; *N. caninum* surface antigen 1 (SAG1, previously named NcP36), and SAG-related sequence protein 2 (SRS2, previously named NcP43). NcSAG1 is anchored on the surface of tachyzoites by glycosylphosphatidylinositol (GPI) anchor [25], while NcSAS2 also has GPI anchor [26], and is expressed in both tachyzoites and bradyzoites [24].

### 4.2. Micronemes

Micronemes are rod-like organelles occurring at the anterior part of the zoites [27]. The proteins produced by micronemes are called MIC or MAP (MIC-associated proteins) [28]. Virtually, this protein family is responsible for the gliding motility of almost all apicomplexan parasites [29]. In *N. caninum*, NcMIC2 [30], NcMIC3 [20], NcMIC4 [31], NcMIC6 [32], NcMIC19, and NcMIC26 [33] can initiate motility, attachment and invasion of the host cells, and apical microneme antigen 1 (AMA1) is a pivotal constituent of the moving junction (MJ) complex and plays an important role in host cell invasion [34,35,36].

### 4.3. Rhoptries

In *T. gondii*, rhoptries are club-shaped apical complex organelles that contain a narrow portion called a rhoptry neck and another part termed the bulb portion [27]. The protein secretion/excretion of rhoptries is called RON (if the protein is secreted from the neck part) (TgRON), while proteins of the rhoptry bulb compartment is termed as ROP as reviewed for *T. gondii* (TgROP) [23]. TgROP proteins are essential for the initial invasion process and in the later stage of invasion, localization, and establishment of intracellular parasitophorous vacuoles (PV) [37]. After attachment to the host plasma membrane, TgRON proteins are excreted, resulting in the formation of the moving junction triggered by TgAMA1 [36]. Numerous ROP proteins of *N. caninum* are identified as potential virulence factors, with some considered active protein kinases as NcROP18 [38], whilst others are deficient in the known kinase-like catalytic domain and are regarded as pseudokinases such as, NcROP2Fam-1, NcROP5, NcROP40 [39], and NcROP16 [40]. Numerous RON proteins of *N. caninum* had been already identified using immunoproteomic analysis such as NcRON4; NcRON5 [41] and NcROP1, 8, 30; and NcRON2, 3, 4, 8 [42].

### 4.4. Dense Granules

The proteins secreted from dense granules are called GRA proteins [43]. Dense granules are small homogenous vesicle-like structures located at both the apical end and behind the nucleus [43]. *Toxoplasma* GRA proteins are secreted at the sub-apical part of the parasite and are targeted to the PV by means of typical signal peptides in the N-terminal part of most GRA proteins [44]. PV acts as a physical and chemical barrier from host defensive molecules, and is vital for the sustainability of infection. GRA proteins are among the most frequently studied secretory proteins not only in *T. gondii*, but also in *N. caninum*. Several orthologues were identified in *N. caninum*, such as GRA1 [45], GRA2 [46], GRA6 [47], GRA7 [48], and GRA14 [49]. In *N. caninum* and other apicomplexan parasites, GRA proteins have appeared to have participated in several vital processes for the parasite’s invasion and maintenance. These processes consist of nutrient acquisition, providing an environment for the parasite growth, proliferation, and egress and of modulating immune response to infection [28,50,51,52].

## 5. Host Range, Transmission, and Life Cycle

*N. caninum* has a heteroxenous life cycle and can infect a wide range of hosts. Two distinct ways of development in the *N. caninum* life cycle have occurred—sexual and asexual reproduction. Sexual reproduction occurrs only in the definitive host in animals of *Canidae* such as the dog [53], coyote [54], grey wolf [55], and dingo [56]. These animals can experience clinical neosporosis, play an important role in the transmission of *N. caninum,* and consequently, are relevant to the epidemiology of the disease. On the other hand, asexual reproduction appears to occur in many intermediate hosts such as cattle. Although most research studies have focused on cattle and dogs as important hosts for *N. caninum*, however, there is increasing evidence that the intermediate host range is wider than anticipated. For instance, *N. caninum* has been recognized in many other animals and may play a role in the epidemiology of neosporosis such as chickens [57], sparrows [58], red foxes [59], and white-tailed deer [60]. The parasite can be transmitted transplacentally in several hosts and a vertical route is the major route of its transmission in cattle. However, horizontal transmission through oocyst ingestion is also a significant transmission route in cattle. Carnivores can acquire infection by the ingestion of infected tissues [1]. Infected dogs or canine animals can shed unsporulated oocysts in their feces [61]. In case of ingestion of sporulated oocysts contaminating pastures or water by susceptible intermediate hosts such as cattle, sporozoites are liberated in the gut; subsequently they invade the gut wall and transform into tachyzoites. Tachyzoite is its rapidly growing stage—residing in PV, it reproduces asexually by endodyogeny. Dissemination of tachyzoites to other types of cells throughout the host via macrophage may occur, including dissemination into the brain, muscle, liver, and lung [62]. In a later stage, tachyzoites differentiate into bradyzoites (slow replicating form by endodyogeny) that form cysts in tissues which can persist throughout the life of the intermediate host [63]. The life cycle is complete when tissue cysts (e.g., in contaminated meat) are ingested by a canine species [53].

## 6. Pathogenesis and Clinical Neosporosis in Different Animals

*N. caninum* is one of the most efficiently transmitted parasites in cattle. The pathogenesis of abortion due to *Neospora* is complex and not fully understood. Bovine neosporosis is mainly a disease of the placenta and fetus, induced by a maternal parasitemia, triggered either via a primary (exogenous) maternal infection or following recrudescence of a persistent (endogenous) infection during pregnancy. Because the parasite is transmitted across the placenta very efficiently and the majority of calves infected in utero are born healthy, it has been questioned whether the parasite is a primary cause of abortion or an opportunistic invader [64,65].

The primary infection of a cow results from the ingestion of sporulated *N. caninum* oocysts [59,66]. Another pathway to infection is the endogenous transplacental transmission of *N. caninum* which is thought to be the most common mode of infection in cattle [67]. Moreover, seropositive cows are more likely to abort than seronegative cows [64]. These findings suggest the reactivation of an already established infection, possibly induced by the immunosuppressive conditions such as those related to mid-gestation [7,68]. In infected cattle, *N. caninum* is usually localized in the central nervous system (CNS) and skeletal muscle [69], presumably to form a tissue cyst containing bradyzoites [70]. Experimentally infected non-pregnant cattle with *N. caninum* do not develop a significant clinical disease. These animals are resistant to infection predominantly due to cell-mediated immune (CMI) mechanisms [71,72], mediated by cytotoxic T lymphocytes. This protection may have persisted into the early stages of pregnancy and vanished at midgestation [68], indicating that pregnancy allows reactivation of tissue cysts of *N. caninum* leading to the release of bradyzoites. At late pregnancy, immune response may restrict parasite transmission and protect the fetus from the detrimental effect of *N. caninum* infection. This protective immunity might be caused by a resumption of the cellular immune response in pregnant dams and the maturation of the immune system of the fetus [7,68].

Following parasitemia, *N. caninum* parasitizes the maternal caruncular septum before crossing to the fetal placental villus [73]. In cases of abortion, a significant degree of damage has to occur in the fetus or its placenta and this effect can be triggered by several pathways. Parasite-induced placental damage may affect fetal survival directly or indirectly through detrimental inflammatory responses, leading to luteolysis and abortion [74]. Another pathway is fetal damage due to primary tissue damage caused by the multiplication of *N. caninum* in the fetus, or due to hypoxia or inanition, secondary to placental damage. Moreover, maternal immune expulsion of the fetus may occur, which is associated with the release of maternal pro-inflammatory cytokines in the placenta.

In cattle, abortion is the major clinical sign of *N. caninum* infection in both beef and dairy cattle [1]. Cows may abort from three months of their pregnancy term, with highest abortion rates existing between five and six months of pregnancy. Fetuses may die in the uterus, resorbed, autolyzed, stillborn, born alive with clinical signs, or born normal but persistently infected. Some *N. caninum*-infected cows may show repeated abortions [75]. Other than abortions in cows, neonatal calves born from an infected dam may exhibit congenital anomalies, neurologic signs, an inability to rise, and a birth weight below the average.

Dogs of all ages are susceptible to *N. caninum* infection. The most complicated cases of neosporosis occur in young, congenitally infected pups. Hind limb paresis that develops into paralysis is commonly reported in young dogs. Forms and signs of neurological disorders are dependent on the site of infection. In the most frequent cases, the hind limbs are more severely affected than the fore limbs. Other less common disorders of neosporosis include difficulty in swallowing, paralysis of the jaw, muscle flaccidity, and necrosis of the skin [76].

In sheep, neosporosis can cause abortion, neonatal mortality, and clinical signs in adult animal. *N. caninum* DNA was detected in the brains of aborted fetuses from infected sheep. Moreover, specific antibodies were detected in the fetuses of aborted ewes. These studies indicate transplacental transmission of *N. caninum* in sheep. In addition, *N. caninum* DNA was found in the brain of an adult encephalitic Merino ewe, suggesting that the parasite might cause clinical neosporosis in adult sheep [77].

In goats, neosporosis can also induce abortion and congenital defects in the fetus, and *N. caninum* DNA was found in aborted fetuses as reviewed by Dubey and Schares, (2011) [78]. In addition, an experimental infection of pregnant goats with *N. caninum* Nc-Spain 7 isolate resulted in fetal death associated with necrosis and a high parasite burden in placentomes and fetal kidney, brain and liver [78]. Not only fetuses, but also adult male goats naturally infected with *N. caninum* showed inflammatory responses in the brain as revealed by infiltration with many immune cells, particularly microglia [79].

Regarding humans, although the infection was found in two rhesus monkeys (*Macaca mulata*) and specific antibodies to *N. caninum* in humans have been recorded, there is no evidence that *N. caninum* infection is zoonotic or has induced clinical forms [1].

## 7. Immunity to *N. caninum*

Both the innate and acquired components of the immune system play important roles against *N. caninum* infection. Although *N. caninum* is an intracellular parasite in mammalian host cells, and therefore requires primarily cellular immune responses, experimental animal models of infection show that a full component of the immune system (cellular and humoral) are critical for the development of optimal resistance to the infection.

### 7.1. Recognition Receptors

#### 7.1.1. Recognition Receptors Description

The protective response against pathogens relies on the proper response of both innate and adaptive immunity. Macrophages and DCs are the main defensive cellular tools of these two arms of immunity. They act as powerful antigen-presenting cells that may elicit effector T cell responses or induce T cells to perform regulatory responses, depending on their activation status. They express pattern recognition receptors (PRRs), that can recognize and react to pathogens and danger signals. Upon activation through their interaction with conserved molecular patterns associated with pathogens (PAMPs), PRRs induce a propagation of signaling cascades that activate pro-inflammatory responses on DCs followed by the differentiation of antigen-specific T cells into protective effector Th1, Th2, and Th17 cells. PRR activation also triggers programmed cell death or apoptosis. However, apoptosis may also trigger the elimination of infectious agents or tumor cells. Therefore, recognition of pathogen- and damage/danger-signals by PRRs is an efficient strategy that links intrinsic cell death programs and complex immune cell interactions to maintain the normalcy of cell microenvironment, leading to protection against infection [80]. In cattle, the majority of work on bovine toll-like receptors (TLRs) concentrates on TLRs 2 and 4. However, homologues of human TLRs 1–10 exist within cattle homologues sharing at least 95% nucleotide sequence identity [81].

TLRs are PRRs that constitute a class of functional, type-I transmembrane glycoproteins that are expressed by various cells ranging from myelomonocytic cells to endothelial and epithelial cells. So far, 12 functional TLRs, subdivided based on cellular location, have been identified in mice (10 in human), which have the ability to interact with microbial PAMPs and host derived damage-associated molecular patterns (DAMPs) to initiate an inflammatory response. TLR1/2/4/5/6 are expressed on the surface of innate cells such as macrophages, DCs, and neutrophils where they recognize microbial antigens, whilst TLR3/7/8/9 are expressed exclusively in intracellular vesicles such as endosomes, where they mainly function in the recognition of microbial nucleic acid [82]. Except for TLR3, the majority of TLRs signal using the adaptor protein Myeloid differentiation primary response 88 (MyD88). Plus, TLR4 utilizes both MyD88-dependent and MyD88-independent pathways. Moreover, stimulation of TLRs via MyD88 activates the Nuclear factor kappa B (NF-κB) and mitogen-activated protein kinase (MAPK) cascades, whereas the MyD88 independent pathway leads to stimulation of the interferon regulatory factor 3/7 (IRF-3/-7) signaling pathway [83].

Nucleotide oligomerization domain (NOD)-like receptors (NLRs) are cytosolic sensors of the innate immune system which interact with distinct molecular motifs followed by modulating the immune response. The deep biological activity of NLR-ligand recognition is linked with the discernible association of polymorphisms in NLR genes [84]. Dectin-1 is a c-type lectin receptor (CLR), and a transmembrane natural killer cell receptor-like C-type lectin presented in many cell types, peculiarly in macrophages, DCs, and neutrophils, suggesting its essential role in immune responses. Dectin-1 participates in the recognition of microbial compounds resulting in several immunological processes such as cytokine production and phagocytosis. Dectin-1 activation has also been shown to enhance TLR mediated activation of NF-κB [85].

The C-C chemokine receptor type 5 (CCR5) receptor is primarily located on T cells, macrophages, dendritic cells, eosinophils, and microglia and it is a G protein–coupled receptor which functions as a chemokine receptor in the C-C chemokine group. CCR5 was found to bind the C-C chemokine macrophage inflammatory proteins-1β (MIP-1β), regulated upon activation, normal T cell expressed and presumably secreted (RANTES), and monocyte chemoattractant protein 2 (MCP-2) via transfected and peripheral blood mononuclear cells [86].

#### 7.1.2. Impacts of *N. caninum* and Its Derived Molecules on Host Recognition Receptors

It has been shown that *N. caninum* tachyzoites or its derived antigens can be efficient at activating many TLRs, suggesting the major role of receptors in modulating *N. caninum* infection. In this regard, Nc-Spain7 and Nc-Spain1H could activate TLR2 expression in bovine trophoblast cell lines [87]. *N. caninum* tachyzoites of different isolates (Nc-1) are also responsible for higher cytokine production such as interleukin-12p40 (IL-12p40), tumor necrosis factor-α (TNF-α), IL-1β, IL-6, and IFN-γ) in bone marrow-derived macrophages (BMDM) of wild-type (WT) TLR2^−/−^ mice [88]. Not only tachyzoites or whole antigens but also a specific derived antigen of *N. caninum* can perform this effect. *N. caninum* glycosylphosphatidylinositol (GPI) has the ability to enhance TLR2 and TLR4, producing different cytokines in the case of macrophages and DC cells of mice [89]. Both cyclophilin (NcCyp), protein disulfide isomerase (NcPDI), rhoptry protein 2 (NcROP2), and 40 (NcROP40) of *N. caninum* appear to interact with TLR2, vaccine antigens of such molecules induce strong protective immunity in mice relied on TLR2 [90,91]. Even in cattle, TLR2 and TLR9, through vaccination with profilin (NcPF) were accompanied by a higher production of systemic IFN-γ than non-vaccinated animals [92]. Added to TLR2, TLR4 is also essential for controlling *N. caninum* infection in many animal models; in infected dogs, one single nucleotide polymorphism was recorded, suggesting the role of TLR4 in canine neosporosis [93]. Consistently, mice deficient in TLR4 showed high sensitivity to *N. caninum* (Nc-1) infection either inoculated intraperitoneally or intragastrically [94,95]. Many other TLRs can also be stimulated by *N. caninum* infection. In C57BL/6 mice, TLR3 and TLR11 are activated by infection of peritoneal and bone marrow-derived macrophages by *N. caninum* Nc-l and Nc-Liv isolates, respectively [96,97,98]. Similarly in cattle, TLRs 3, 7, 8, 9 are also activated at the materno–fetal interface either by *N. caninum* tachyzoites [99], or by inactivated soluble whole antigens or recombinant NcSAG1, NcHSP20 or NcGRA7 [100].

CCR5 is proficiently activated and plays a role in *N. caninum* infection as shown from mouse model-based studies. NcCyp and excreted/secreted antigens induced recruitment of murine monocytes to the site of infection via Gi protein and CCR5-dependent pathways [101]. Recombinant NcCyp caused the CCR5-dependent migration of murine and bovine cells [102]. Moreover, using a CCR5^−/−^ mouse model, infected knockout (KO) mice showed higher mortality and clinical disturbance than WT mice; this was associated with poor migration of DCs and NKT cells in the later mouse type [103].

NOD2 is a cytosolic receptor that colocalized to PV. Infection of mouse BMDM by *N. caninum* (Nc-1) resulted in lower proinflammatory cytokine (IL-6 and TNF-α) and higher anti-inflammatory cytokine (IL10) in NOD2^−/−^ than WT ones. Interestingly, in vivo studies revealed that mice lacking NOD2 showed a higher resistance to infection than WT mice [104]. This phenomenon was also noticed in another study with a different receptor target; Dectin 1^−/−^ mice exhibited higher resistance after infection with *N. caninum* (Nc-1 isolate) which may be related to a higher production of IL-12p40 in various immune cells in the same mouse strain [105]. The *N. caninum* infection in mouse macrophages upregulated the activity of peroxisome proliferator-activated receptor gamma (PPAR-γ) which resulted in the promotion of M2 macrophage phenotypes (alternatively activated macrophages), especially when associated with NF-*κ*B inhibition [106].

The inflammasome is a multiprotein intracellular complex that detects pathogenic microorganisms and exerts a protective function via an induction of pro-inflammatory cytokines IL-1β and IL-18 productions. In particular, NLRP3 inflammasome in peritoneal macrophages or BMDM of mouse origin appears to be activated by infection with *N. caninum* tachyzoites (Nc-1). This effect was associated with the release of IL-1β and IL-18, as well as cleavage of caspase-1 and cell death [107,108]. A similar effect was also reported when a cattle macrophage cell line was infected by tachyzoite of Nc-1 [109]. From these studies, we can conclude that *N. caninum* tachyzoites or different derived molecules can efficiently manipulate host cell receptors, but the effect is not evidently similar and ends in a higher resistance or susceptibility to infection depending on the type of host, receptor, or effector immune cell.

### 7.2. Signaling Transduction Pathways

#### 7.2.1. Signaling Pathways Description

The nuclear factor kappa B (NF-κB) signaling pathway is involved in the regulatory machinery of several physiological and pathological conditions, including immune responses. The NF-κB family consists of five members: p65 (RelA), p100/p52, p105/p50, c-Rel, and RelB. NF-κB proteins form homo- or heterodimeric structures upon activation. The canonical NF-κB pathway includes the phosphorylation of the inhibitory IκB proteins by the IκB kinase complex (IKK), which leads to the ubiquitination and further degradation of IκB, leading to nuclear translocation of NF-κB dimers and activation of κB-responsive target genes. The canonical pathway is stimulated by a wide range of stimuli, such as TNF-*α* and IL-1β [110]. On the other hand, a non-canonical NF-κB pathway is detected in a more cell-specific fashion including immune-related cells and lymphoid tissue. The non-canonical pathway is activated by specific stimuli that include B cell-activating factor (BAFF) and Lymphotoxin-β (LTβ) [111].

Nuclear factor of activated T cell (NFAT) proteins were first identified in T-cells as transcriptional activators of IL-2, a key regulator of T cell immune response [112]. There are four types of proteins in the NFAT gene family: NFATc1, NFATc2, NFATc3, and NFATc4. Such proteins are mediated by the phosphatase calcineurin that dephosphorylates NFAT proteins to expose their nuclear localization signals, thus inducing the nuclear translocation of NFAT proteins. Subsequently, in the nucleus, NFAT proteins interact with other factors to trigger the target gene expression, essential for many biological functions [113]. The phosphoinositide 3-kinase/AKT (PI3K/AKT) pathway is an important signal transduction pathway that links several classes of membrane receptors to many vital cell processes, such as cell survival, proliferation, and differentiation [114]. After activation of PI3K, these molecules can induce recruitment and activation of the serine/threonine-specific protein kinase AKT through phosphorylation-induced activation of transmembrane phosphatidylinositol (4,5) bisphosphate (PIP_2_) into phosphatidylinositol (3,4,5) trisphosphate (PIP_3_). Activated AKT can subsequently phosphorylate and activate several other proteins. Virtually, AKT’s action induces and regulates a large array of cellular processes [115,116].

The Janus Kinase/signal transducers and activators of transcription (JAK/STAT) pathway is considered an important membrane-to-nucleus cascade, which may be stimulated by a large scale of stimuli such as cytokines, reactive oxygen species, and growth factors. JAK/STAT is involved in the normal development and homeostasis of cells, and in the control of cell proliferation, differentiation, cell migration, and apoptosis [117]. Moreover, this pathway appears to regulate many vital processes such hematopoiesis, immune response, and adipogenesis [118]. The signaling activation is induced by the specific binding of inducers (e.g., IL-6) to initiate the oligomerization of respective receptor subunits (e.g., cytokine receptors), leading to signal enhancement by phosphorylation of the receptor-associated tyrosine kinases JAK1-3 and Tyk2 [119]. In a particular way, JAK activation proceeds after the receptor subunit comes into close proximity and allows the cross-phosphorylation of these tyrosine kinases. Then, activated JAKs induce the phosphorylation of the receptor that now serves as a docking site for other JAK targets involving their major substrates known as signal transducer and activator transcription factors (STATs). STAT proteins have the dual effect of signal transduction and transcription activation downstream of phosphorylation processes. Virtually, STAT phosphorylation induces the dimerization of other STATs, culminating with a translocation to the nucleus mediated by importin α-5. Inside the nucleus, the dimerized STATs bind to specific regulatory sequences along the DNA, leading to activation or repression of target genes [120].

Mitogen-activated protein kinases (MAPKs) includes a family of kinases that have a main role in tumor growth and metastasis. MAPKs can be divided into three subfamilies: the extracellular-signal-regulated kinases (ERKs), the c-Jun N-terminal kinases (JNKs), and p38 MAPKs that, together with the JNKs, form the stress-activated protein kinase pathways [121]. All MAPKs have special links to the regulation of intracellular metabolism, gene expression, cell growth and differentiation, apoptosis, stress response, and immune responses [122].

Caspases (cysteine-aspartic protease, cysteine-aspartases or cysteine-dependent aspartate-directed proteases) are a family of protease enzymes playing essential roles in programmed cell death and inflammation. Caspase-1, 4, 5, 11 are involved in inflammatory responses [123]. Caspase-1 is key factor in activating pro-inflammatory cytokines and for immune cell recruitment to the site of damage. Caspase-1 therefore plays a fundamental role in the innate immune system. The enzyme is responsible for processing cytokines such as IL1-β and IL18 [123].

The l-arginine-nitric oxide pathway is a regulating mechanism triggered by the formation of nitric oxide (NO) from the amino acid l-arginine by the nitric oxide synthase. NO is produced abundantly in the tissues of cardiovascular and nervous systems. In all these tissues, the l-arginine-NO pathway acts as a transduction mechanism for the soluble guanylate cyclase regulated by NO. In addition, NO can be secreted from numerous immune cells such as macrophages, neutrophils, and other cells, suggesting its role in the host defense mechanism either against tumor cells or invasive organisms [124,125].

#### 7.2.2. Major Signaling Pathways and *N. caninum* Infection

Accumulating evidence had indicated that the NF-κB signaling pathway is a major cascade in manipulating most protozoan infections including *N. caninum*. NcGRA6 could activate the NF-κB pathway using a luciferase reporter assay and 293T cells transfected with the NcGRA6 gene [126]. Furthermore, *N. caninum* 14-3-3 protein is defined as one of the key molecules of the parasite responsible for the activation of the NF-κB pathway. Treatment of peritoneal macrophages harvested from C57BL/6 mice with recombinant Nc14-3-3 resulted in the activation of NF-κB/p65 [127]. This effect might be regulated by the involvement of MyD88 adaptor protein, because it is also reported to be activated in cases of treatment of murine peritoneal macrophage or BMDM with soluble tachyzoite antigens. The implementation of an in vivo study using MyD88^−/−^ mice infected with *N. caninum* (Nc-1) showed a higher susceptibility and a higher parasite burden as well as lower IL-12 and delayed IFN-γ productions [128,129].

Another effective pathway during *N. caninum* infection is the MAPK pathway, which is proceeded by several pathways including ERK, p38MAPK, and JUNK proteins as reported in an earlier section in this review. There are several reports that correlate one or more MAPK pathways with *N. caninum* infection or derived molecules using different animal models. In mice, treatment of peritoneal macrophages with recombinant 14-3-3 protein activated the MAPK and AKT pathways accompanied with an increase in IL-6, IL-12p40, and TNF-α [127]. Moreover, *N. caninum* infection in BMDM of mice, enhanced the production of above-mentioned cytokines (IL-6, IL-12p40, TNF-α) in addition to IFN-γ and IL-1β in C57BL/6 mice mediated by the MAPK pathway [88]. Jin et al. (2017) [97] also reported that the infection of peritoneal macrophages of mice resulted in activation of the MEK-ERK pathway which is also mediated by the TLR11 pathway. In addition, p38 MAPK was phosphorylated in mouse BMDM after stimulation with live or antigen extract of *N. caninum* (Nc-1) isolate [130]. Even in ruminant animal models—in cattle and in goats—activation of macrophages or neutrophils p38 MAPK and ERK1/2- with *N. caninum* live tachyzoites triggered the formation of extracellular trap-like structures (ETs) [131,132,133]. Furthermore, in dogs, *N. caninum* live tachyzoite is associated with the activation of the p38 MAPK/ERK/1/2 pathway in neutrophils, resulting in ET formation [134]. In another study, the PI3K pathway was involved in ET formation in caprine neutrophils [135].

NcGRA7 stimulated efficiently the NFAT pathway using an in vitro transfection system of 293T cells. This effect was also associated in cases of mice infection with NcGRA7^−/−^ parasites with lower parasite virulence and IFN-γ levels in ascites fluid [126]. Moreover, NcROP16 secretion in human fibroblast cells resulted in activation of STAT3 and nuclear translocation. This effect was associated in vivo with reduced parasite growth when KO parasite infected in BALB/c mice [40]. *N. caninum* live tachyzoite is also associated with an activation of different pathways in different animals such as JAK-STAT pathways in bovine monocytes [136], and l-arginine pathways in murine macrophages [137]. This section supports the efficiency of *N. caninum* or its specific derived molecule for modulating the host factors at signaling transduction pathways, and subsequently unravels its professionalism in invading many animal species. A deep understanding of differential mechanisms of *N. caninum*–host cell interactions at the molecular interface will greatly assist in developing control strategies against neosporosis. Indeed, abundant previous reports into this research field has provided valuable knowledge on *N. caninum* interactions with recognition receptors or signaling pathways. This knowledge has already been exploited to develop potent vaccines against neosporosis as reported by several studies using mice [90,91] and cattle [100]. *N. caninum*–host interaction at the recognition receptor and signaling pathway level is summarized in Table 1.

### 7.3. Cellular Immune Responses

#### 7.3.1. General Description of Effector Cellular Immune Compartments

Macrophages or monocytes are one of the most important cells that contribute to innate immunity to almost all pathogens. They are capable of influencing the adaptive immune response directly through antigen presentation or indirectly by secreting many effector molecules including cytokines. During infection, macrophages have been shown to contribute to parasite clearance via the process of phagocytosis, as well as through the production of proinflammatory cytokines and NO [138].

Cluster of differentiation 4^+^ (CD4^+^) T cells contribute to the resistance by releasing cytokines that regulate other innate and adaptive immune cells, and by activating B cells to ensure the efficient production of specific antibody responses to parasite antigens. CD4^+^ T helper cells provide signals that regulate B cells survival and differentiation into antibody-producing cells [139].

IFN-γ is a type II interferon, which is a cytokine important for innate and adaptive immunity against protozoal and other intracellular pathogens. IFN-γ is produced primarily by natural killer (NK) and natural killer T (NKT) cells as part of the innate immune response, and by CD4^+^ Th1 and CD8^+^ cytotoxic T lymphocyte (CTL) effector T cells after antigen-specific immune response develops. It exerts an essential function by the activation of macrophages and major histocompatibility complex (MHC) class II molecule expression. The importance of IFN-γ in the immune system is in part due its ability to inhibit pathogen replication directly, and most importantly from its immunoregulatory effects. Cellular responses to IFN-γ are induced through its interaction with a heterodimeric receptor consisting of interferon gamma receptor 1 (IFNGR1) and interferon gamma receptor 2 (IFNGR2) [140].

#### 7.3.2. Cellular Immunity and *N. caninum* Infection

##### Effector Immune Cells

Two main subsets of immune cells have been studied and showed great significance in protection against *N. caninum*—macrophages and T lymphocytes. As a first line of the immune defense mechanism, macrophages are reported as a critical element for combating *N. caninum* infection. In mice, a common experimental model of neosporosis conducted by Abe et al. (2014) [141] noticed that *N. caninum* infection in C57BL/6J mice induced significant macrophage recruitment at the site of infection. Furthermore, targeted depletion of macrophages in mice with clodronate lipososmes intraperitoneal injection resulted in high susceptibility of mice against infection. Moreover, Fereig et al. (2019) [142] found that NcGRA6 recombinant antigen could stimulate macrophage in vitro to secrete IL-12p40 in a dose-dependent manner. In the same context, macrophage and monocyte function has been elucidated in cattle—the most important susceptible animals to neosporosis. In vitro infection of bovine macrophages with *N. caninum* (Nc-Spain7 and Nc-SpainH1 isolates) is associated with an elevation of the parasite gene expression responsible for regulatory immune response and inflammatory processes from the host [143]. Furthermore, in a similar study investigating the same host cell and parasite isolate, elevated levels of reactive oxygen species (ROS) and IL-12 was recorded in Nc-SpainH1 than Nc-Spain7 infected and non-infected cells. In addition, infected macrophages with both isolates showed lower expression of MHC Class II, CD86, and CD1b molecules than non-infected cells [144]. In another study, bovine monocytes infected with Nc-Liv isolate produced higher levels of IL-1β [136]. Thus, macrophages of different phenotypes exerted similar immune responses against *N. caninum* infection either in murine or bovine cell origin.

Dendritic cells also revealed a proficient immune response against *N. caninum* tachyzoites or its derived molecules in a way that enhances combat against the infection. Exposure of BMDMs to *N. caninum* infection (Nc-1) induced higher levels of IL-12 and TNF-α, and additionally IFN-γ in C57BL/6 and BALB/c mice, respectively [145,146]. In addition, the treatment of T cells with viable tachyzoites or with antigenic extracts of Nc-1 induced IFN-γ production when co-cultured with DCs but not with macrophages, which were stimulated using antigenic extract only [147]. DCs contributed to the control of *N. caninum* infection because in early infected mice, IL-12 production was markedly noticed in both conventional and plasmacytoid DCs of C57BL/6 mice [148].

Regarding T-cells, many subsets have also been shown to affect *N. caninum* infection in various hosts. Both CD4^+^ (helper) and CD8^+^ (cytotoxic) T cells are determinant factors for the outcome of neosporosis [149]. In mice, CD4^+^ or CD8^+^ cells have been strongly correlated to control the infection mostly via the IFN-γ-dependent pathway [148,149]. In cattle, a similar effect also has been reported. Orozco et al. (2013) [150] found that *N. caninum* seropositive uteri showed higher number of lymphocytes (CD4^+^ and CD8^+^) than seronegative pregnant cows. Similar findings have been reported in *N. caninum*-infected cows under natural field conditions [151,152]. In addition, in vitro stimulation of bovine CD4^+^ cells with *N. caninum* antigens triggered the production of IFN-γ [71]. Moreover, *N. caninum* water soluble lysate screened for immunopotency using CD4^+^ cells revealed NcSAG1, NcSRS2, NcGRA2, NcMIC3, NcGRA7, and NcMIC11 as candidates for stimulation of CD4^+^ cells [153]. CD3^+^ and MHC II^+^ cells were detected in the bovine fetal tissues infected with *N. caninum* Nc-6 isolate [154]. In BALB/c athymic nude mice that were lacking thymus, infection with *N. caninum* (JPA1), isolate was associated with a high severity of infection in comparison to WT mice [155].

NKT and NK cells also have been established as cellular components of a control strategy against *N. caninum* either in mice [156] or cattle [157]. In cattle also, infection of peripheral blood mononuclear cells (PBMCs) either with *N. caninum* Nc-Spain7 or Nc-Spain8 induced IFN-γ and IL-4 productions started from 6 dpi [158]. Regarding B-cells—the important source of antibody production—infected B-cell deficient mice showed higher sensitivity to *N. caninum* (Nc-1) infection. Moreover, stimulation of spleen cells from WT mice with *N. caninum* antigens produced a higher level of IFN-γ than that of B-cell deficient mice [159]. The summary of cellular immune response for effector cells is illustrated in Table 2.

##### Effector Immune Molecules

Cytokines, ROS and reactive nitrogen species (RNS) are considered to be important factors in the pathogenesis of *N. caninum* infections. These compounds are secreted by the host as a response to control the proliferation and dissemination of the parasite. The type and degree of the response depends on several factors related to the host or the parasite, or even on the experimental approach.

IFN-γ is considered the life-saving molecule for the infected host with *N. caninum* and also other intracellular parasites, triggering a protective immune response by inducing a predominant CD4^+^ Th1 cell polarization. As explained earlier, NK, NKT, CD4^+^ Th1, and CD8^+^ cytotoxic T lymphocyte are the main producing cells for IFN-γ secretion. In cases of *N. caninum* infection, the role of IFN-γ is the most studied molecule for unraveling strategy to control the parasite. These studies have been conducted using different animal models, cell types and approaches. In murine model, infection of C57BL/6 mice with *N. caninum* (Nc-1) revealed a high response of IFN-γ production or gene expression and also higher susceptibility in mice lacking such a molecule or relevant mediators [160,161,162,163]. In BALB/c mice; another commonly used mouse model in *N. caninum* research, the acutely infected mice showed higher levels of IFN-γ than non-infected mice and this effect is correlated for the protective effect against infection in vitro or in vivo [164,165,166,167]. The role of IFN-γ has been investigated also in CBA/Ca swiss white and A/J mice, as well as in rats and also showed its critical role in infection control [168,169,170]. In the fat*-*tailed dunnart (*Sminthopsis crassicaudata*)—a mouse-like marsupial animal—spleen cells infected with *N. caninum* (Nowra isolate) exhibited higher expression levels of IFN-γ, associated with a similar effect in other cytokines such as TNF-α, IL-4, and IL17A [171]. Not only in the murine model, the role of IFN-γ has been investigated also in different animal models either in experimental or natural field conditions. Bovine cells, tissues or blood from infected cattle showed a greater increase in IFN-γ protein production or gene expression than those of non-infected animals. This effect was slightly variable in levels but similar in outcomes that resulted in *N. caninum* control [151,152,172,173,174,175,176,177,178,179,180]. In the same context in sheep, the treatment of ovine fibroblasts with recombinant IFN-γ greatly induced the reduction of parasite growth compared to non-treated cells [181]. Similarly in dogs, the presence of IFN-γ in Madin-Darby canine kidney (MDCK) cells resulted in a low viability of infected cells and an increase in apoptotic cell death [182,183]. These results indicated the secretion of IFN-γ in various cells of different hosts in a response against *N. caninum* infection regardless of used isolates. Thus, this cytokine is encountered as a vital molecule for host resistance against *N. caninum* infection.

However, IL-12 is another critical cytokine for the control of *N. caninum*, which is primarily related to the IFN-γ production mechanism also. Cao et al. (2018) [184] found that treatment of DCs with a secretory protein dynein LC8 light chain 2 (NcDYNLL2) resulted in a higher production of IL-12 and also IFN-γ in BALB/c mice. A similar effect was also reported for IL-12 production in the same mouse strain and immune cells (DCs) after infection with tachyzoites of *N. caninum* (Nc-1) isolate [185]. Many reports also indicated that *N. caninum* or its derived molecules can trigger the stimulation of proinflammatory cytokines and mediators other than IFN-γ and IL-12 in different animal models such as inducible nitric oxide synthase 2 (iNOS2) and NO in mice and rats [186,187,188] and TNF-α, IL-8, and IL-17 in cattle [87,189,190]. Delayed hypersensitivity immune reaction in skin tissues was also reported in naturally or experimentally infected cattle [191]. In naturally infected goats, *N. caninum* increased the level of NO but did not advance oxidation protein products [192]. Thus, proinflammatory cytokines and oxidative mediators are critical for controlling the neosporosis via its direct destructive effects on the parasite.

Not only proinflammatory cytokines, but also anti-inflammatory ones may play an essential role in regulating immunity against *N. caninum* infection, particularly IL-4 and regulatory IL-10. IL-10 has a potent anti-inflammatory effect which favors the tissue regeneration in infected hosts; moreover, it is associated with increased parasite proliferation. In most studies, an increase of IL-10 is associated with an increase of some proinflammatory cytokines; this may require more studies targeting IL-10 profiles during the course of *N. caninum* infection. In cattle, IL-10 was reported to increase in fetal and maternal tissues [173,177,178,179]. In Qs mice, Quinn et al. (2004) [193] found that IL-4 was exclusively increased in cases of infected/pregnant mice rather than infected/non-pregnant mice while IFN-γ, TNF-α, and IL-12 increased in infected pregnant and non-pregnant mice, suggesting its role in maintaining pregnancy during *N. caninum* infection. However, IL-4 neutralization during infection of pregnant mice did affect congenital transmission to offspring [194]. Moreover, Nishikawa et al. (2003) [167] found that the balance between IL-4/IFN-γ is important for protective immunity against *N. caninum* infection using BALB/c mice.

In a number of studies using different immune cells from various animal models, *N. caninum* infection showed high efficiency in triggering extracellular traps like the specific networks limiting parasite proliferation and mediating the killing of parasites. This effect was noticed in cases of bovine neutrophils [131], caprine monocytes [133], and canine neutrophils [134]. Indoleamine 2,3-Dioxygenase (IDO) is reported to participate in the control of *N. caninum* infection through tryptophan degradation-dependent pathways and mediated by IFN-γ [186,188,195,196,197].

Several studies have found that bioactive molecules can mediate the host defense against *N. caninum* infection, and in some cases ameliorate the fatal consequences of neosporosis. These biomolecules are gaining wide attention for their anti-*N. caninum* activities. Perforin and granzyme are potent effector molecules released from cytotoxic T cells and are participated in the killing of *N. caninum* infection in cattle [198]. A marked change in a group of host factors at the feto–maternal interface in cows has been detected in several studies. Lectin-binding architectural patterns were disrupted by *N. caninum* (Nc-1) infection [199]. Reduction of pregnancy associated proteins 1 and 2 in pregnant cattle have been associated with a higher rate of *Neospora*-induced abortion in cows [200]. *N. caninum* infection in cattle has been associated with a decrease in SERPINA14 levels in pregnant cows resulting in abortion. SERPINA14 has a regulatory decrease of the immunosuppressive effect of progesterone in cases of pregnancy [201,202]. Although there is an unknown immunological role, host factors such as neurotropic proteins in rat tissues [203], cholesterol and lipoproteins in human and bovine cell lines [204], purine, acetylcholinesterase and butyrylcholinesterase in gerbil tissues and blood [205,206] seem to play a role in *N. caninum* infection in mentioned hosts because of marked altered levels among infected and non-infected hosts. Furthermore, a previous report indicated that cathelicidins (host defence peptides in human cells) showed higher expression levels in macrophages infected with live tachyzoites of *N. caninum* than naïve cells. This elevation was associated with increments in TNF-α, IL-1β, IL-8, IFN-γ cytokine and reduced parasite internalization in naïve macrophages [207]. The summary of cellular immune response for effector molecules is illustrated in Table 3.

### 7.4. Humoral Immune Responses

Humoral immunity or antibody-based immunity is mediated by antibodies, complement proteins and certain anti-microbial peptides. In cases of *N. caninum* as well as most intracellular pathogens, the role of humoral immunity is transient but cannot be neglected. Regarding complement-mediated killing, the role in cases of *Neospora* and even in *T. gondii* is poorly understood. In cases of *T. gondii*, it actively resists complement-mediated killing in non-immune human serum (NHS) by inactivating C3b [208]. On the other hand, there are numerous studies that revealed specific antibody production either in natural or experimental infection with *N. caninum* in various animal models.

*N. caninum* dense granule proteins are the most extensively investigated molecules as antigens capable of detection of specific antibodies in various hosts, but surface-derived molecules showed the highest reactivity for antibody detection. Nevertheless, whole tachyzoite lysate antigens (NLA) are still widely used and are considered as a standard antigen in evaluating new diagnostic candidates. In cases of cattle, specific antibodies to NcSAG1, NcSRS2, NcGRA6, and NcGRA7 [209,210,211,212,213] were detected. Furthermore, specific antibodies to NLA, especially IgG1 and IgG2, were detected in sera of infected cattle [214,215,216]. While in dogs, specific antibody to NcSAG1, NcGRA2, NcGRA6, NcGRA7, and NcPF was detected [210,217,218,219]. In sheep also, specific antibodies against *N. caninum* infection was detected as evidenced by high levels of NcGRA7 in experimentally or naturally infected animals [220]. Specific antibodies against NcSAG1, NcGRA6, and NcGRA7 in BALB/c mice [213], and NcGPI in Swiss OF1 mice [89] were detected. Nishikawa et al. (2000) [221] reported that in vitro treatment of Hs68 cells of human origin with monoclonal antibodies against *N. caninum* tachyzoites resulted in significant inhibition of parasite invasion and growth in the host cells. This inhibition might be induced by surface-related antigens from the parasites. However, transferring antibodies against *N. caninum* did not protect the pregnant mice after challenging with the parasites in mid-gestation. The study revealed that the transferred antibodies disappeared during pregnancy upon parasite infection [222]. A summary of specific antibody production (total IgG and its subtypes or IgM) against *N. caninum* or its specific molecules is summarized in Table 4.

## 8. Concluding Remarks

In the last decade, an abundance of research reports related to the recognition or signaling pathways that interact with *N. caninum* have been conducted. An increasing knowledge of the crosstalk between different receptors or signaling cascades among *N. caninum* tachyzoite or derived molecules and many factors of various animal hosts ultimately promotes the understanding of pathogenesis and the impact of neosporosis. In fact, many recognition receptors and interacting pathways are currently reported or being tested in animal models, most of them acting as a weapon to combat the infection. Signaling pathways are responsible for the outcome of immune responses. This response might be mediated by an interaction of parasite-derived molecules to the transcription factors either directly (parasite inside cell) or indirectly (parasite outside cells) (Figure 1).

Nevertheless, specific roles of these pathways and the microenvironment accompanied *N. caninum* infection in different hosts, particularly in cattle, dogs and mice, is still unknown. Hence, it is critical to perform more research studies and pursue a more complete understanding of the cascade-dependent signals which lead to immune response, consequently this will lead to the development of fully effective control strategies against *N. caninum* in susceptible hosts including potent vaccines or effective therapeutic agents.

## Figures and Tables

**Figure 1 pathogens-09-00384-f001:**
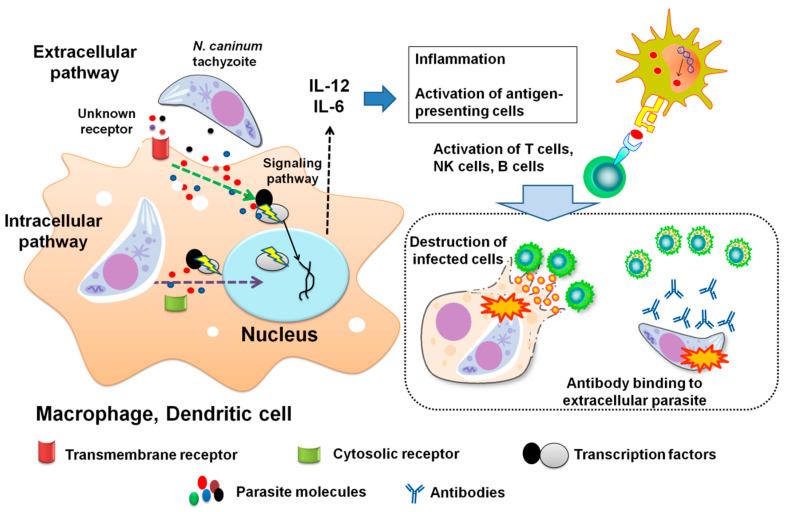
Possible pathways of interaction of *Neospora caninum* tachyzoite or derived molecules and host immune effectors. Cases of infection with *N. caninum* tachyzoites or parasite molecules (e.g., NcGPI, NcGRA6, NcGRA7, or NcCyp), are uptaken by macrophages or dendritic cells as professional antigen-presenting cells. This recognition is mediated by recognition receptors that are either transmembrane like most toll-like receptors (TLR) or cytosolic like nucleotide oligomerization domain (NOD)-like receptors. Afterwards, activation of host signaling pathways such as NF-kB is occurred to stimulate immune responses. In cases of *N. caninum* infection, proinflammatory cytokines as interleukin (IL)-6 and IL-12 are secreted, which subsequently enhance other immune cells to perform more potent and specialized effects. T-helper cells exert this role by producing IFN-γ which is a key molecule in combating *N. caninum* infection. Damage of the infected cells can be performed by cytotoxic T-cells or natural killer cells. Production of specific antibodies by B cells also plays a role in relieving subsequent infection by antibody binding to extracellular parasites.

**Table 1 pathogens-09-00384-t001:** Recognition and signaling pathways contributed to host cell–*N. caninum* crosstalk.

Host Factors	Host Species	Parasite or Its Molecule	Impacts and Outcomes	References
TLR2	Bovine trophoblast and caruncular cells	*N. caninum* tachyzoites (Nc-Spain7 and Nc-Spain1H)	Higher mRNA expression levels of TLR-2 were noticed in the trophoblast cell line infected with the low-virulence Nc-Spain1H.	[87]
TLR2 and MAPK	BMDM from C57BL/6 (WT) and TLR2^−/−^ mice	*N. caninum* tachyzoites (Nc-1)	*N. caninum* extracellular vesicles significantly increased the production of IL-12p40, TNF-α, IL-1β, IL-6, and IFN-γ by WT-BMDMs than in TLR2^−/−^ mouse BMDMs mediated by MAPK signaling pathway.	[88]
TLR2 and TLR4	Mouse macrophages cell line and DCs from OF1 mice	NcGPI	NcGPI induced stimulation of TLR2 and TLR4 from HEK cells, and TNF-α, IL-1β and IL-12 secretion by macrophages and DCs. NcGPIs reduced expression of MHC molecules of class I on DCs.	[89]
TLR2	DCs and in vivo assays in BALB/c mice	NcPDI, NcROP2 NcROP40 (Nc-Spain7)	Vaccination of mice with cocktail antigen mixed with OprI; TLR2 adjuvant induced a Th1/Th2 immune response in adult mice and conferred protection in adult and offspring mice. In vitro, cocktail antigens stimulated secretion of TNF-α in DCs.	[90]
TLR2	Spleen cells from C57BL/6 mice and TLR2^−/−^ mice and in vivo assay	NcCyp-entrapped with oligomannose-coated liposomes (Nc-1)	Immunized WT mice with NcCyp-OML showed high protection against *N. caninum* infection in comparison to TLR2^−/−^ immunized mice. Spleen cells from immunized WT mice with NcCyp-OML showed higher IFN-γlevels than those of TLR2^−/−^ mice.	[91]
TLRs 2 and 9	Bovine plasma	NcPF (Nc-1)	The vaccine formulated with TLRs 2 and 9 agonists improved the production of systemic IFN-γ and induced long-term recall B-cell responses.	[92]
TLR4	Dog blood	*N. caninum* in naturally infected animals	In genotyped sample, one *TLR4* SNP marker was recorded in seropositive dog samples for *N. caninum*.	[93]
TLR4- and IL-12Rβ2	WT and 57BL/10ScCr mice lacking TLR4 and IL-12 receptors	*N. caninum* tachyzoites (Nc-1)	All C57BL/10ScCr mice but not WT were succumbed by 8 dpi. KO mice showed higher parasite burden in the internal organs than WT controls, which might be correlated with reduced IFN-γ and increased IL-4 expressions.	[94]
TLR4- and IL-12Rβ2	WT and 57BL/10ScCr mice lacking TLR4 and IL-12 receptors	*N. caninum* tachyzoites (Nc-1)	TLR4^−/−^ and IL-12Rβ2^−/−^ were succumbed after intragastric challenge with *N. caninum* tachyzoites. In contrast, WT-BALB/c mice challenged with parasites remained alive for at least 6 months.	[95]
TLR3 and TRIF	BMDM from WT and TLR3^−/−^ and TRIF^−/−^ mice	*N. caninum* tachyzoites (Nc-Liv and Nc-1)	Infection of macrophages from mice with targeted deletions in various innate sensing genes demonstrates that host responses to *N. caninum* are dependent on the TLR3 and the adapter protein TRIF. RNA from *Neospora* elicited TLR3-dependent type I IFN responses.	[96]
TLR11 and ERK	Peritoneal macrophages from C57BL/6 mice	*N. caninum* tachyzoites (Nc-1)	*N. caninum* infection rapidly activated MEK-ERK signaling via TLR11 in mouse peritoneal macrophages. *N. caninum* infection elevated IL-12p40 by macrophages, which was significantly reduced via inhibition of TLR11/MEK/ERK pathway.	[97]
TLR3 and TRIF	BMDM from C57BL/6 and TLR3^−/−^ and TRIF^−/−^	*N. caninum* tachyzoites (Nc-Liv)	TLR3^−/−^ and TRIF^−/−^ mice showed higher parasite burdens, increased inflammatory lesions, and reduced production of IL-12p40,TNF-α, IFN-γ, and NO. *N. caninum* tachyzoites and RNA recruited TLR3 to the PV and translocated IRF3 to the nucleus, and upregulated the expression of TRIF in murine macrophages.	[98]
TLR3, 7 and 8, 9	Bovine placenta and fetal spleen.	*N. caninum* tachyzoites (Nc-1)	mRNA expression levels of TLRs 3, 7, 8, and 9 were high in the spleen of fetuses from *N. caninum*-infected heifers, and in the placenta and maternal caruncle from infected heifers.	[99]
TLR3, 7, 8 and 9	Fetal-maternal interface of cattle	Inactivated soluble whole antigens (Nc-6), and rNcSAG1, rNcHSP20 and rNcGRA7 (Nc-1)	Heifers immunized with inactivated soluble antigens and recombinant NcSAG1, NcHSP20 and NcGRA7 showed higher TLR7 and 8 expressions in caruncles than non-immunized heifers.	[100]
CCR5	Peritoneal monocytes and BMDM from C57BL/6 and CCR5^−/−^	*N. caninum* tachyzoites excreted/secreted antigens (Nc-1)	Excreted/secreted antigens from *N. caninum* (NcESAs) attracted monocytes to the site of infection in both in vitro and in vivo. NcCyp in the NcESAs might work as chemokine-like proteins and NcESA-induced chemoattraction involved CCR5 contribution.	[101]
CCR5	Murine and bovine cells	NcCyp (Nc-1)	Recombinant protein of NcCyp induces the migration of murine and bovine cells in a CCR5-dependent manner	[102]
CCR5	DC and NKT cells from C57BL/6 J and CCR5^−/−^ mice and in vivo assays.	*N. caninum* tachyzoites (Nc-1)	In the *N. caninum*-infected CCR5^−/−^ mice, increased mortality and neurological dysfunctions, poor migration of DCs and NKT cells to the site of infection were observed. Higher IFN-γ and CCL5 levels were associated with brain tissue damage of CCR5^−/−^ mice during the infection, and a primary microglia culture from CCR5^−/−^ mice showed lower IL-6 and IL-12 productions against *N. caninum*.	[103]
NOD2	BMDM and in vivo assay using C57BL/6, (NOD2^−/−^) mice	*N. caninum* tachyzoites (Nc-1)	Infection of macrophages with *N. caninum* increased expression of NOD2, and NOD2^−/−^ macrophages decreased IL-6 and TNF-α, and increased production of arginase-1 and IL-10 In vivo, NOD2^−/−^ mice reduced MAPK phosphorylation and IL-6 production, and decreased inflammation in organs with higher parasite burden, but mice were partially resistant to lethal doses of tachyzoites.	[104]
Dectin-1	BMDM, DCs and spleen homogenate from C57BL/6 and Dectin-1^−/−^	*N. caninum* tachyzoites (Nc-1)	Lacking Dectin-1 rescued 50% of the mice infected with *N. caninum,* and Dectin-1^−/−^ mice presented a reduction in the parasite load during acute and chronic phases. In vitro, IL-12p40 increased in *N. caninum* infected macrophages, DC and spleen cells of Dectin-1^−/−^ mice than WT.	[105]
PPAR-γ	BALB/c mice	*N. caninum* tachyzoites (Nc-1)	In vitro study, *N. caninum* treated macrophages revealed promotion of M2-ploarized phenotype compared with the GW9662 (PPAR-γ inhibitor) group and RGZ (PPAR-γ agonist) group, through up-regulating the activity of PPAR-γ and inhibiting NF-κB activation.	[106]
NLRP3 inflammasome	Peritoneal macrophages from C57BL/6 and Nlrp3^−/−^ mice	*N. caninum* tachyzoites (Nc-1)	Inflammasome activation-mediated caspase-1 processing and IL-1β cleavage in response to infection with *N. caninum* were observed and correlated with the time of infection and infective dose.	[107]
NLRP3 inflammasome	BMDM from C57BL/6 and Nlrp3^−/−^ mice	*N. caninum* tachyzoites (Nc-1)	In vitro results showed that *N. caninum* infection of murine BMDMs activated the NLRP3 inflammasome, associated with the release of IL-1β and IL-18, cleavage of caspase-1, and induction of cell death. Infection of *Nlrp3*^−/−^ and caspase-1/11^−/−^ mice resulted in decreased production of IL-18 and reduced IFN-γ in serum.	[108]
Inflammasome mediate-caspase-1	Cattle macrophage cell line	*N. caninum* tachyzoites (Nc-1)	Inflammasome-mediated activation of caspase-1 occurs in *N. caninum*-infected bovine macrophages. Caspase-1-dependent cell death triggered in *N. caninum*-infected cells.	[109]
NF-kB	293T human cell lines	NcGRA6 (Nc-1)	293T cells were transfected with the luciferase reporter plasmids and the expression vector of NcGRA6 gene encoding protein. Cells transfected with NcGRA6 gene strongly activated NF-kB.	[126]
MAPK, AKT, and NF-kB	Peritoneal macrophages from C57BL/6 mice	Nc14-3-3 (Nc-1)	Recombinant Nc14-3-3 activates the MAPK and AKT signaling pathways, associated with an increase of IL-6, IL-12p40, and TNF-α. Phosphorylated NF-kB/p65 was observed in peritoneal macrophages treated with rNc14-3-3.	[127]
MyD88	C57BL/6 and MyD88^−/−^ mice	*N. caninum* tachyzoites (Nc-1)	Sub-lethal infection induced acute mortality of MyD88^−/−^ mice. Higher parasite burden in MyD88^−/−^ mice was associated with reduced IL-12 production by DCs, delayed IFN-γ responses by NKT, CD4^+^ and CD8^+^ T lymphocytes, and production of high levels of IL-10.	[128]
TLR2/MyD88	Various immune cells from C57BL/6 b, TLR2^−/−^ and MyD88^−/−^ and in vivo assay	*N. caninum* soluble antigen and tachyzoite (Nc-1)	Peritoneal macrophages and BMDDC exposed to *N. caninum*-soluble antigens increased the expression of TLR2. In case of infection, CD4^+^ and CD8^+^, and IFN-γ:IL-10 ratio decreased, and parasite burden increased in TLR2^−/−^ mice than WT mice.	[129]
MAPK	BMDM from C57BL/6 and in vivo assay	*N. caninum* tachyzoites (Nc-1)	p38 phosphorylation wasinduced in macrophages stimulated by live tachyzoites and antigen extracts, while its inhibition increased IL-12p40production. In vivo blockade of p38 increased production of cytokines, enhanced survival against the infection.	[130]
ERK 1/2- and p38 MAPK	Bovine neutrophils	*N. caninum* tachyzoites (Nc-1)	*N. caninum* tachyzoites triggered extracellular trap formation in bovine neutrophils via ERK 1/2-, or p38 MAPK-signaling pathway	[131]
ERK 1/2- and p38 MAPK	Bovine macrophage	*N. caninum* tachyzoites (Nc-1)	*N. caninum* tachyzoites induced bovine macrophage-derived extracellular trap-like structures, which may be mediated by an ERK 1/2- and p38 MAPK-pathway.	[132]
ERK 1/2- and p38 MAPK	Caprine monocytes	*N. caninum* tachyzoites (Nc-1)	*N. caninum* tachyzoites triggered extracellular trap formation in caprine monocytes by ERK 1/2-, or p38 MAPK-signaling pathway dependent manner.	[133]
ERK 1/2, and p38 MAPK	Canine neutrophils	*N. caninum* tachyzoites (Nc-1)	*N. caninum* tachyzoites strongly induced NETs formation in canine neutrophils ERK 1/2, and p38 MAPK signaling pathways.	[134]
PI3K	Caprine neutrophils	*N. caninum* tachyzoites (Nc-1)	The inhibition of PMN autophagy via inhibition of the PI3K mediated signaling pathway resulted in failure of tachyzoite-induced NETosis.	[135]
NFAT	293T cells transfected with the luciferase reporter plasmids	NcGRA7 (Nc-1)	Infection with *NcGRA7*^−/−^ parasites showed reduced virulence in mice. The levels of IFN-γ in the ascites fluid, CXCL10 expression in the peritoneal cells, and CCL2 expression in the spleen were lower 5 dpi with the *NcGRA7*^−/−^ parasite than the parental strain.	[126]
STAT3	HFF cells and in vivo assay using BALB/c mice.	NcROP16 (Nc-1)	NcROP16 secretion in host cell phosphorylates STAT3, and pSTAT3 then migrates to the cell nucleus. Deletion of NcROP16 decreased parasite growth kinetics in vitro and reduced virulence in mice.	[40]
JAK-STAT	Bovine monocytes	*N. caninum* tachyzoites (Nc-Liv)	Neonatal monocytes are more resistant to cellular invasion with *N. caninum* and the magnitude of the responses is related to significant changes in the JAK-STAT pathway.	[136]
l-arginine/NO	BALB/c and NO^−/−^ mice	*N. caninum* tachyzoites (Nc-1)	Production of NO increased in cultures of macrophages treated with IFN-γ, and dose-dependent growth inhibition was observed. Blockade of l-arginine-dependent pathway, NG-monomethyl-l-arginine, reduced the inhibitory effects induced by IFN-γ.	[137]

**Table 2 pathogens-09-00384-t002:** Cellular immune response against *N. caninum* involving various immune cells.

Host Factors	Host Species	Parasite or Its Molecule	Impacts and Outcomes	References
Macrophages	C57BL/6J mice	*N. caninum* tachyzoites (Nc-1)	Marked increase in recruitment of macrophages to site of infection associated with increased IL-6 and IL-12p40 during *N. caninum* infection. Macrophage-depleted mice exhibited high susceptibility to *N. caninum* infection.	[141]
Macrophages	BALB/c mice	NcGRA6 (Nc-1)	Peritoneal macrophages treated with different doses of recombinant NcGRA6 induced cytokine production IL-12.	[142]
Macrophages	Cattle	*N. caninum* tachyzoites (Nc-Spain7 and Nc-Spain1H)	Infection of cattle accompanied with higher genes expression involved in pathogen recognition, chemotaxis and proinflammatory and regulatory cytokine secretion.	[143]
Macrophages	Cattle	*N. caninum* tachyzoites (Nc-Spain7 and Nc-Spain1H)	Macrophages infected with Nc-Spain1H showed high ROS production and IL12p40 expression, compared to cells infected with Nc-Spain7. IL-10 was increased in macrophages infected with both isolates. Infected macrophages exhibited lower expression of MHC Class II, CD86, and CD1b molecules than uninfected cells.	[144]
Monocytes	Cattle	*N. caninum* tachyzoites (Nc-Liv)	Neonatal animals had a marked higher percentage of CD14^+^ monocytes, and adult monocytes showed higher parasitism than neonatal monocytes. Greater secretion of IL-1β was observed in neonates than adult monocytes.	[136]
Dendritic cells	C57BL/6 mice	*N. caninum* tachyzoites (Nc-1)	Cytokine expression analysis revealed that both viable and nonviable parasites stimulated BMDCs to express IL-12p40, IL-10, and TNF-α.	[145]
Dendritic cells	BALB/c mice	*N. caninum* tachyzoites (Nc-1)	The response to whole tachyzoites (live, heat-killed, freeze-killed) or whole-cell tachyzoite lysate (soluble, insoluble antigen) stimulated moderate-to-high levels of IL-12, IFN-γ, and TNF-α.	[146]
Dendritic cells and macrophages	Murine H ^2k^ cell line and CBA/J mice	*N. caninum* tachyzoites (Nc-1)	IFN-γ-increased in T cells co-cultured with DCs exposed to viable tachyzoites or antigenic extract. Oppositely, IFN-γ production triggered after interactions between T cells and macrophages exposed to antigenic extract only.	[147]
T cells and dendritic cells	C57BL/6 mice	*N. caninum* tachyzoites (Nc-1)	In early infection, IL-12 production by conventional and plasmacytoid DCs was increased in mesenteric lymph nodes. Increased proportions and numbers of TCRαβ+CD8+IFN-γ+ lymphocytes were detected, not only in the intestinal cells and lymph nodes, but also in the spleen of the infected mice.	[148]
CD4^+^ and CD8^+^ T cells	BALB/c mice	*N. caninum* tachyzoites (Nc-1)	Most of the anti-CD4 mAb-treated mice and all of the anti-CD4 and anti-CD8 mAbs-treated mice died within 30 dpi. In contrast, 100% of PBS-treated mice and 70% of anti-CD8 mAb-treated mice survived more than 30 days.	[149]
CD4^+^ and CD8^+^ T cells	Heifers	*N. caninum* in naturally infected animals	More lymphocytes were observed in the uteri of the seropositive pregnant animals than in the seronegative pregnant animals. CD4^+^ and to lower extent CD8+ cells were distributed in the endometrium and myometrium of the non-pregnant cows and were sparse in the placentomes of pregnant cows.	[150]
T cells	Bovine mononuclear cells from peripheral blood	*N. caninum* tachyzoites (Nc-Liv)	Percentages of CD2^+^ and CD4^+^ T-cells in PBMC increased after infection in both early and late gestation, Percentages of CD8^+^ T-cells increased 1–2 wpi at day 70.	[151]
CD4^+^ and CD8^+^ T cells	Heifer	*N. caninum* in naturally infected animals	An infiltration of CD4^+^ and CD8^+^ T cells were significantly increased.	[152]
CD4^+^ T cells	Cattle	*N. caninum* tachyzoites antigen (Nc-1)	The concentration of bovine IFN-γ in supernatant collected from CD4^+^ T cells stimulated with *Neospora* antigen fractions was higher than samples incubated with mock.	[71]
CD4^+^ T cells	Cattle	*N. caninum* tachyzoites water soluble lysate (NcWSA) (Nc-1)	NcWSA was fractionated by HPLC and screened for immune-potency using CD4^+^ve T cell lines. The approach revealed six target proteins (SAG1 SRS2, GRA2, MIC3, GRA7, and MIC11).	[153]
MHC II^+^ and CD3^+^ cells	Bovine fetal tissue	*N. caninum* tachyzoites (Nc-6)	The immunolabeling of MHC II^+^ and CD3^+^ cells in fetal tissues was associated with fetal infection with *N. caninum* Nc-6 Argentina isolate.	[154]
Thymus	BALB/c nu/nu (athymic nude) and BALB/ c (WT) mice	*N. caninum* tachyzoites (Japanese isolate JPA1)	All the athymic nude mice died within 28 days after intraperitoneal injection of tachyzoites, whereas all the WT mice survived without exhibiting any clinical signs. Tachyzoites were identified in the uterus and pancreas and later spread to many other organs, and nude mice developed high level of serum IFN-γ and IL-6 as infection proceeded.	[155]
Natural killer T cells	BALB/c mice	*N. caninum* tachyzoites (Nc-1)	The parasite burden in the brain of mice was promoted by the treatment depletion of NKT in mice.	[156]
Natural killer and CD8^+^ cells	Bovine peripheral blood lymphocytes	*N. caninum* tachyzoites (Nc-Liv)	Phenotyping of peripheral blood lymphocytes showed a drop in the NK cells at 4–6 dpi, followed by an increase in both NK cells and CD8^+^ T cells at days 11–15. A whole blood flow cytometric assay showed that CD4^+^ T cells were the major IFN-γ producing cells, but in the early infection both NK cells and CD8^+^ T cells contributed to IFN-γ production.	[157]
Peripheral blood mononuclear cells	Cattle	*N. caninum* tachyzoites *N. caninum* tachyzoites (Nc-Spain7 and Nc-Spain8)	In vitro stimulation of PBMCs from heifers of both infected groups triggered a significant increase of IFN-γ and to a lower extent IL-4 levels from 6 dpi onwards than non-infected group.	[158]
B-cells	C57BL/6(WT) and B-cell^−/−^ µMT mice	*N. caninum* tachyzoite and antigen (Nc-1)	WT mice were resistant to disease, but µMT mice died starting from 29 dpi onwards. Tachyzoite antigen-stimulated spleen cells of both WT and µMT mice infected with *N. caninum* showed a marked proliferative depression at 10 dpi. At 24 dpi, this immunosuppression was still maintained in µMT mice whereas it was restored in WT mice. Stimulated splenocytes of infected µMT mice secreted significantly less IFN-γ and less IL-10 than corresponding WT splenocytes.	[159]

**Table 3 pathogens-09-00384-t003:** Cellular immune response against *N. caninum* involving immune effector molecules.

Host Factors	Host Species	Parasite or Its Molecule	Impacts and Outcomes	References
IFN-γ	C57BL/6 (WT) and various KO mouse strains	*N. caninum* tachyzoites (Nc-1)	CD8^−/−^ mice infected with *N. caninum* showed higher parasitic loads in the brain and lungs than WT ones. Mice treated with IFN-γ-expressing CD8^+^ T cells showed lower parasitic burdens than IFN-γ-deficient CD8^+^ T cells.	[160]
IFN-γ and TNF-α	C57BL/6 mice	*N. caninum* tachyzoites (Nc-1)	Expression levels of IFN-γ and TNF-α were high in the brains of infected mice. The level of neurotransmitters glutamate, glycine, gamma-aminobutyric acid, dopamine and 5-hydroxytryptamine were altered in infected mice.	[161]
IFN-γ and leptin	C57BL/6 mice	*N. caninum* tachyzoites (Nc-1)	In early infection, parasites were detected in the adipose tissue associated with increased numbers of immune cells, and increased expression of IFN-γ in gonadal adipose tissue. In chronic cases, parasite DNA was not detected in these tissues, but Th1 cell numbers remained above control levels, and marked increase of serum leptin was detected.	[162]
IFN-γ	C57BL/6 mice	*N. caninum* tachyzoites (Nc-1)	NK cells and various T cell populations mediate production of IFN-γ in the adipose tissue of *N. caninum* infected mice.	[163]
IFN-γ and IL-12	BALB/c mice	*N. caninum* tachyzoites (Nc-1)	Mice treated with anti-IFN-γ alone increased morbidity/mortality, and increased foci of liver necrosis and increased parasite numbers in the lung by 7 dpi. Mice treated with rIL-12 decreased encephalitis and brain parasite load at 3 wpi.	[164]
IFN-γ	BALB/c and IFN-γ^−/−^ mice	*N. caninum* tachyzoites (Nc-1)	Infected IFN-γ^−/−^ mice died earlier than WT. IFN-γ^−/−^ mice failed to increase MHC class II expression on macrophages. BALB/c mice induced T-cell proliferation while IFN-γ^−/−^ mice did not. In serum, high levels of IFN-γ and IL-4 were detected in resistant hosts, whereas IL-10 was detected in IFN-γ^−/−^ mice. The levels of IL-12 in IFN-γ^−/−^ mice were higher than in BALB/c mice at 7 dpi.	[165]
IFN-γ	BALB/3T3 clone A31 mice	*N. caninum* tachyzoites (Nc-1)	The viability of *N. caninum*-infected murine fibroblast cells was significantly reduced after treatment with mouse IFN-γ. FasL expression was clearly induced by *N. caninum*-infection and IFN-γ- treatment, and the reduction in host-cell viability was prevented with the addition of anti-mouse FasL monoclonal antibody (mAb).	[166]
IFN-γ and IL-4	BALB/c and IFN-γ^−/−^ mice	Tachyzoites and SRS2 antigen *N. caninum* (Nc-1)	In the acute infection of *N. caninum*, IFN-γ^−/−^ mice showed high levels of IL-10 production, whereas significant levels of IFN-γ and IL-4 production were observed in resistant WT mice. BALB/c mice vaccinated with a virus expressing NcSRS2 were protected against parasite and low levels of IFN-γ and high levels of IL-4 productions were observed.	[167]
IFN-γ and IL-12	A/J mice	*N. caninum* tachyzoite and antigen (Nc-1)	Inbred A/J mice developed no clinical and little histologic evidence of infection by *N. caninum*. Splenocytes obtained from infected mice proliferate in vitro in response to both *N. caninum*-soluble antigens. Mice infected with *N. caninum* produce significant quantities of IL-12 and IFN-γ.	[168]
IFN-γ	CBA/Ca and Swiss white (BK: W) mice	*N. caninum* tachyzoites (Nc-1)	Infected spleen cells had the highest specific lymphoproliferative response, and a mixed cytokine response with elevated IFN-γ and fairly low IL-4 and IL-10 secretion was recorded.	[169]
IFN-γ, TNF-α, IL-10 and TGF-β,	Rat	*N. caninum* tachyzoites (Nc-1)	Treatment of primary mixed cultures of rat astrocytes and microglia with either IFN-γ or TNF-α reduced infection rate. In the absence of IL-10 and TGF-β, tachyzoite numbers were reduced significantly against non-treated cells.	[170]
IFN-γ, TNF-α, IL-17A and IL-4	Fat-tailed dunnart	*N. caninum* tachyzoites (Nc-Nowra)	mRNA expression during the time course of infection revealed higher levels of IFN-γ, TNF-α, IL-17A, and IL-4 cytokines in infected rather than non-infected dunnart spleen cells.	[171]
IFN-γ and TNF-α	Cattle	*N. caninum* tachyzoites (Nc-1)	Addition of recombinant IFN-γ in primary astroglia-microglia culture inhibited the tachyzoite growth, which was not reversed by the addition of an NO antagonist. TNF-α, to a lesser extent, also inhibited the tachyzoite growth.	[172]
IFN-γ, IL-12, TNF-α, IL-10	Heifers	*N. caninum* tachyzoites(Illinois cattle isolate)	Infected dams showed an increased number of lymphocyte subpopulations compared with uninfected pregnant animals. Gene expression increased both Th1 and Th2 cytokines in *N. caninum*-infected animals (IFN-γ, IL-12, TNF-α, IL-10).	[173]
IFN-γ	Cattle	*N. caninum* tachyzoites (Nc-Liv)	The live and heat-inactivated tachyzoites of *N. caninum*, directly trigger production of IFN-γ from purified and IL-2-activated bovine NK cells.	[174]
IFN-γ and IL-4	Cattle	*N. caninum* tachyzoites (Nc-Liv)	Percentages of CD2^+^ and CD4^+^ T-cells in peripheral blood mononuclear cells (PBMC) increased after infection in both early and late gestation. Percentages of CD8^+^ T-cells increased 1–2 wpi at day 70. IL-4 and IFN-γ mRNA expression in PBMC increased 1–2 wpi at day 210 and IL-4 increased 1–2 wpi at day 70.	[151]
IL-2, IFN-γ, IL-12p40, TNF-α, IL-18, IL-10, and IL-4	Heifer	*N. caninum* tachyzoites (Nc-Liv)	Infection in early gestation induced an increase in mRNA levels of IL-2, IFN-γ, IL-12p40, TNF-α, IL-18, IL-10, and IL-4 in placental tissues. This was associated with extensive placental necrosis and an infiltration of CD4^+^ T cells and macrophages, and IFN-γ and TGF-β was also highly and moderately increased, respectively.	[175]
IFN-γ and IL-17	Cattle	*N. caninum* tachyzoites (Nc-Liv)	Naive T-cells in contact with infected macrophages produced both IFN-γ and IL-17 in a pattern that is dependent on whether the priming macrophage was protected or non-protected.	[176]
IFN-γ, IL-4, IL-12p40, IL-10, TNF-α and MHC class II	Heifer	*N. caninum*-naturally infected animals	An increase in IFN-γ and IL-4 mRNA was detected. IL-12p40, IL-10, and TNF-α were also significantly increased. MHC Class II antigens were expressed on maternal and fetal epithelial and stromal fibroblastoid cells.	[152]
IFN-γ, TNF-α, IL-12p40, IL-10, and IL-4	Heifer	*N. caninum*-naturally infected animals	*N. caninum* infected dams showed elevated mRNA levels of IFN-γ, TNF-α and IL-12p40, and IL-10, but expression of IL-4 did not differ significantly among the groups.	[177]
IFN-γ, IL12, TNF-α, IL-6, IL-10 and IL-4	Heifer	*N. caninum* tachyzoites (Illinois cattle isolate)	Fetuses had higher expression of most cytokines at 3 and 9 wpi in fetuses that were alive at 6 wpi. In dams, most cytokines were down-regulated from 6 wpi, with elevated IL-4 expression observed in the caruncle.	[178]
IFN-γ, IL-10 and IL-4	Heifer	*N. caninum* tachyzoites (Nc-Spain7)	In infected dams with live fetuses, IFN- γ increased in both caruncle and cotyledon, and IL-10 elevated in cotyledon. Infected live fetuses showed elevated expression of IFN-γ and IL-10 in fetal spleen, and showed diminished expression of IL-4 in the thymus compared to control uninfected fetuses.	[179]
IFN-γ, IL-17 and IL-4	Heifer	*N. caninum* infection (Nc-Spain7)	In dams, significantly higher IFN-γ and IL-4 levels were found in the experimentally infected animals compared to the control or naturally infected dams. IL-17A production was very low in the dams infected with *N. caninum* and did not seem to be a major regulator of IFN-γ production in this model.	[180]
IFN-γ	Sheep	*N. caninum* tachyzoites (Nc-1 and Nc-Liv)	The *N. caninum* Nc-1 multiplied more quickly in fibroblast cells than the Nc-Liv isolate. Treatment of the cells with ovine rIFN-γ for 24h before infection inhibited intracellular multiplication of the parasite.	[181]
IFN-α, IFN-β and IFN-γ	Dog	*N. caninum* tachyzoites (Nc-1)	IFN-γ inhibited the parasite growth in MDCK cells infected with *N. caninum* tachyzoites with greater efficacy than IFN-α and IFN-β.	[182]
IFN-γ	Dog and BALB/c cell lines	*N. caninum* tachyzoites (Nc-1)	In the presence of IFN-γ, the viability of the infected host cell was decreased and apoptotic cell death occurred. An increase of FasL expression on the IFN-γ-treated cells following *N*. *caninum* infection was observed. IFN-γ treatment decreased *Bcl-2* expression in the cells cultured with *N*. *caninum* while parasite infection increased *Bcl-2*.	[183]
TNF-α and IL-12	BALB/c mice	*N. caninum* cytoplasmic dynein LC8 light chain (NcDYNLL) protein (Nc-1)	NcDYNLL2 is a secretory protein and was internalized by the host immune cells and stimulated TNF-α and IL-12 production by murine dendritic cells.	[184]
IL-12	BALB/c mice	*N. caninum* tachyzoites (Nc-1)	The number of splenic conventional dendritic cells (cDCs) increased at 5 dpi, while the number of plasmacytoid dendritic cells (pDCs) did not change on infection, this effect is associated with upregulation of costimulatory and MHC class II molecules. This stimulatory effect was more marked at the earliest assessed time point after infection, 12 h, when a clear increase in the frequency of cDCs and pDCs producing IL-12 was also observed.	[185]
IFN-γ,TNFR2, IL-10, beta 2 microglobulin (b2M), and inducible nitric oxide synthase (iNOS2)	Various KO and WT mice	*N. caninum* tachyzoites (virulent Nc-1 or Nc-2 or attenuated Ncts-8)	Infection with Nc-1 was 100% lethal in IFN-γ^−/−^ mice. TNFR2^−/−^ and b2M^−/−^ mice were infected with Nc-1 or Nc-2 isolate, while TNFR2^−/−^ or b2M^−/−^ mice were resistant to Ncts-8 infection. Lack of mortality and minimal histopathology scores demonstrated that Nc-1, Nc-2, and Ncts-8 infections were avirulent in IL-10 and iNOS2^−/−^ mice.	[186]
TNF-α and NO	Rat	*N. caninum* tachyzoites (Nc-Bahia isolate)	Astrocytes responded to infection by producing the pro-inflammatory cytokine TNF-α and the neurotoxic-free radical NO, 24, and 72 hpi.	[187]
iNOS	C57BL/6 mice and iNOS^−/−^ mice	*Neospora* tachyzoite (Nc-1)	*N. caninum* infection in WT mice induced NO production in vivo and in vitro, and iNOS^−/−^ mice succumbed during acute infection, associated with increased in parasite load. Infected iNOS^−/−^ mice showed marked CNS inflammation, and increased production of IL-12, IFN-γ, IL-6, TNF-α, and IL-17A.	[188]
GRO-a, IL-8 and IP-10, MCP-1, RANTES, GM-CSF, COX-2 and iNOS	Cattle	*N. caninum* tachyzoites (Nc-1)	*N. caninum* infection in the bovine umbilical vein endothelial cells (BUVEC) in vitro resulted in elevation of the CXC chemokines GRO-a, IL-8, and IP-10, the CC chemokines MCP-1 and RANTES and of GM-CSF, COX-2, and iNOS after 2–4 hpi.	[189]
IL-17	Cattle	*N. caninum* tachyzoites (Nc-Liv)	Infection with the *N. caninum* induced fibroblasts to secrete pro-IL-17 factors by inducing a γΔ17 phenotype that preferentially kills infected target cells.	[190]
TNF-α and IL-8	Cattle	*N. caninum* tachyzoites (Nc-Spain7 and Nc-Spain1H)	TNF-α and IL-8 were elevated, while IL-6 and TGF-β1 were decreased in both trophoblast and caruncular cell lines. Higher secretion of TNF-α were noticed in the trophoblast cell line infected with the low-virulence Nc-Spain1H.	[87]
Delayed type hypersensitivity (DTH)	Cattle	*N. caninum* infection followed by intradermal inoculation of soluble tachyzoite antigen (Nc-1)	Either experimental or natural infection of cows with live *N. caninum* tachyzoites developed skin reactions compatible with DTH between 24 and 96 hpi associated with an increase in IFN-γ release.	[191]
NO and advanced oxidation protein products (AOPP)	Goat	*N. caninum*-naturally infected animals	*N. caninum* seropositive animals showed higher serum levels of NO compared to seronegative animals. AOPP levels did not change in infected and non-infected groups.	[192]
IL-4	Qs mice	*N. caninum* tachyzoites (Nc-Liv and Nc-SweB1)	Spleen cells from both infected/non-pregnant mice produced higher IFN-γ, IL-12, and TNF-α than in infected/pregnant mice. IL-4 was exclusively increased in infected/pregnant mice and thus appear to be responsible for the observed decline in Th1 cytokine production in pregnant mice.	[193]
IL-4	BALB/c mice	*N. caninum* tachyzoites (Nc-1)	In naïve mice before pregnancy, neutralization of IL-4 during gestational challenge did not result in decreased congenital transmission, while in mice that were primed before pregnancy, congenital transmission was significantly decreased. Decreased congenital transmission was associated with significantly lower levels of maternal splenocyte IL-4 secretion, lower IL-4 mRNA levels, and higher levels of IFN-γ secretion.	[194]
Neutrophil extracellular trap	Cattle	*N. caninum* tachyzoites (Nc-1)	*N. caninum* tachyzoites triggered NETosis in a time- and dose-dependent manner. NET structures are released by bovine PMN and entrapping tachyzoites.	[131]
Monocyte extracellular trap (ETs)	Goat	*N. caninum* tachyzoites (Nc-1)	*N. caninum* tachyzoites were not only capable of triggering ETs formation in caprine monocytes, but also that monocyte-released ETs were able to entrap viable tachyzoites.	[133]
Neutrophils extracellular traps (NETs)	Dog	*N. caninum* tachyzoites (Nc-1)	*N. caninum* tachyzoite induced NETs formation as observed by scanning electron microscopy.	[134]
Indoleamine 2,3-Dioxygenase (IDO)	Cattle	*N. caninum* tachyzoites (Nc-1)	Induction of the tryptophan-degrading enzyme indoleamine 2,3-dioxygenase (IDO) inhibited the parasite growth that is mediated by IFN-γ-activated bovine fibroblasts and endothelial cells.	[195]
MIF, TGF-β, IFN-γ and IDO	Human cell lines	*N. caninum* tachyzoites (Nc-1)	*N. caninum* infection increased the macrophage migration inhibitory factor (MIF), mainly in HeLa cells. Conversely, parasite infection induced down-regulation of TGF-β, mostly in BeWo cells. HeLa cells were more susceptible to *Neospora* infection than BeWo cells mediated by IFN-γ via IDO-dependent pathways in HeLa cells alone.	[196]
iNOS, IDO and COX-2	Rats	*N. caninum* tachyzoites (Nc-Bahia)	iNOS, IDO, and COX-2 control the proliferation of *N. caninum* in vitro, while the release of IL-10 by glia affects the inflammation and maintains the parasitism.	[197]
Perforin and Granzyme	Cattle	*N. caninum* tachyzoites (Nc-1)	Enrichment and blocking of CD4^+^ and CD8^+^ T-lymphocyte effector subsets indicated that CD4^+^ CTL killed *N. caninum*-infected, autologous target cells and that killing was mediated through a perforin/granzyme pathway.	[198]
Lectin	Heifers	*N. caninum* tachyzoites (Nc-1)	Changes in the lectin-binding pattern were noted in infected animals as noticed in the glycocalyx, apical cytoplasm of endometrial cells, and apical cytoplasm of the trophoblastic cells.	[199]
Pregnancy-associated glycoprotein; PAG-1, PAG-2	Heifers	*N. caninum* tachyzoites (Nc-Spain7)	Non-aborting infected heifers showed a temporary fall in PAG-1 and PAG-2 concentrations from 7 to 14 dpi. Dams aborting at 14 and 21 dpi showed dramatic PAG-1 and PAG-2 decreases from 14 dpi to undetectable levels upon euthanasia.	[200]
SERPINA14	Heifers	*N. caninum* tachyzoites (Nc-Spain7)	Normally, uterine serpins (SERPINA14) regulate the immunosuppressive effect of progesterone during late pregnancy. *N. caninum* infection downregulates the uterine immunosuppressive function of SERPINA14.	[201]
SERPINA14	Heifers	*N. caninum* tachyzoites (Nc-Spain7)	Intercaruncular *SERPINA14* expression was negatively correlated with *IFN-*γ expression in cotyledon samples and with *IL4* expression in uterine lymph nodes.	[202]
Neurotrophic factors GFAP, BDNF and NGF and cytokine IL-10,	Rat	*N. caninum* tachyzoites (Nc-1)	*N. caninum-*infected glial cultures responded with astrogliosis increased GFAP, IL-10, BDNF, and NGF gene expression.	[203]
Host nutrient molecules	Human and bovine cell lines	*N. caninum* tachyzoites (Nc-Liv)	*N. caninum* utilizes plasma lipoproteins, scavenges cholesterol from Niemann–Pick type C protein 1 -containing endocytic organelles, and salvages sphingolipids from host Golgi Rab14 vesicles.	[204]
Acetylcholinesterase (AChE) and butyrylcholinesterase (BChE)	Gerbils	*N. caninum* tachyzoites (Nc-1)	On 7 dpi a decrease of AChE in total blood and brain was observed, along with reduction of BChE in plasma of infected animals compared to noninfected. AChE activity increased in total blood and reduced in brain at 30 dpi, and BChE activity was markedly increased at 30 dpi.	[205]
Purine	Gerbils	*N. caninum* tachyzoites (Nc-1)	The purine levels (ATP, ADP, AMP, adenosine, inosine and xanthine) in the brain are markedly reduced at 7 dpi, while the purine levels were significantly increased on days 15 (ATP, AMP, adenosine, hypoxanthine, and xanthine) and on 30 PI (ATP, ADP, AMP, adenosine, and uric acid).	[206]
Cathelicidin	Human cell line	*N. caninum* tachyzoites (Nc-1)	Infection of macrophages (THPI) with live tachyzoites elevated cathelicidins were associated with increased pro-inflammatory cytokines (TNFα, IL-1β, IL-8, IFN-γ). This immune response in infected macrophages was mediated by (MEK 1/2) and resulted in reduced parasite internalization in naïve macrophages.	[207]

**Table 4 pathogens-09-00384-t004:** *N. caninum* antigens and humoral immune response.

*Neospora* Antigens	Type of Antibody Response	Hosts	Experimental Procedures	References
NcSAG1 and NcGRA7	Total IgG	Cattle	Sera from aborting cows in the field using iELISA.	[210]
NcGRA6	Various	Cattle and dogs	NcGRA6-based LATEX agglutination test showed high efficacy in distinguishing infected and non-infected bovine and canine sera.	[211]
NcGRA7	IgG	Cattle and Buffaloes	Specific antibody against NcGRA7 was detected in sera of cattle and buffaloes naturally infected with Nc-1 isolate.	[212]
NcSRS2	IgG	Cattle	A blocking enzyme-linked immunosorbent assay (b-ELISA) based on NcSRS2 recombinant protein and polyclonal antibodies against rNcSRS2 (b-ELISA/rNcSRS2) showed high potency for detection of *N. caninum* antibody in cattle.	[213]
NcSAG1, NcGRA6 and NcGRA7	IgG1 and IgG2a	Cattle and BALB/c mice	Sera from acute and chronic infection of experimentally infected mice and sera from aborting cows in the field by Nc-1 using iELISA.	[214]
*N. caninum* tachyzoite BPA1 isolate	Various	Cattle	Experimentally infected cows revealed a predominant IgG2 response in two cows, a mixed IgG1-IgG2 response in two other cows and a predominant IgG1 response in one cow. All five fetuses of infected dams at 9 wpi mounted a strong IgG1 response.	[215]
Soluble tachyzoite antigen	IgG1 and IgG2	Cattle	Sera from crossbred and purebred cows showed changeable antibody levels according to breeds and stages of pregnancy.	[216]
Soluble tachyzoite antigen	IgG1, IgG2 and IgM	Cattle	Higher IgG1 than IgG2 serum levels in the presence of IFN-γ in non-aborting chronically infected cattle reduced vertical transmission of *N. caninum* under field conditions.	[217]
Profilin	IgG1 and IgG2a	Mice and dogs	Anti-NcPF antibodies reflect parasite activation and neurological symptoms in mice and dogs.	[218]
NcGRA2 and NcSAG1	IgG	Dog	In experimentally infected dogs, anti-NcGRA2t antibodies were detected in the acute stage, while anti-NcSAG1t antibodies were detected in both the acute and chronic stages.	[219]
NcGRA7	IgG1 and IgG2	Dog	Specific antibodies were detected in sera of experimentally infected dogs intravenously by Nc-1 tachyzoites at 28 dpi.	[220]
NcGRA7	IgG	Sheep	Sera from experimental and naturally infected aborting and non-aborting sheep using a time-resolved fluorescence immunoassay.	[221]
NcGPI	IgG	Swiss OF1 mice	Anti-NcGPI antibodies were detected by dot blot using NcGPI antigens extracted from tachyzoites of *N. caninum*.	[89]

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
