# Peer review of "From Signaling Pathways to Distinct Immune Responses: Key Factors for Establishing or Combating *Neospora caninum* Infection in Different Susceptible Hosts"

_pathogens, 2020, doi:10.3390/pathogens9050384_

Round 1

Reviewer 1 Report

Reviewer’s Comment on manuscript ID: pathogens-776393

Title: From signaling pathways to distinct immune responses: Key factors for establishing or combating Neospora caninum infection in different susceptible hosts

Comments to the authors

The topic of the manuscript is relevant and original because is the first review about molecular mechanisms and host signaling pathways related to immune responses in Neospora caninum infection. Authors have conducted a very deep and complete review, providing interesting data. Therefore, this manuscript could be of interest to the readership of Pathogens.

However, some concerns should be addressed to improve the quality of the manuscript:

The first sections of the manuscript about general comments on bovine neosporosis are extremely long. I recommend reducing in length the following sections: “Historical view and evolution of Neospora caninum”; “Host range, transmission, and life cycle” and “Pathogenesis and clinical neosporosis in different animals”.

  • No references for the “Overview of recent researches on machinery of neosporosis” section have been included.
  • L33: delete “stormy”.
  • L36-37: Please, include “variables depending on the parasite”.

Several studies that have provided important results about the topic of the present review have not been cited by the authors. I would encourage the authors to consider the following studies:

  • Strohbusch, M., Muller, N., Hemphill, A., Margos, M., Grandgirard, D., Leib, S., et al. (2009). Neospora caninum and bone marrow-derived dendritic cells: parasite survival, proliferation, and induction of cytokine expression. Parasite Immunol. 31, 366-72
  • Feng, X., Zhang, N., and Tuo, W. (2010). Neospora caninum tachyzoite- and antigen-stimulated cytokine production by bone marrow-derived dendritic cells and spleen cells of naive BALB/c mice. J.Parasitol. 96, 717-23
  • Dion, S., Germon, S., Guiton, R., Ducournau, C., and Dimier-Poisson, I. (2011). Functional activation of T cells by dendritic cells and macrophages exposed to the intracellular parasite Neospora caninum. Int.J.Parasitol. 41, 685-95.
  • Regidor-Cerrillo J, Arranz-Solís D, Benavides J, Gómez-Bautista M, Castro-Hermida JA, Mezo M, Pérez V, Ortega-Mora LM, González-Warleta M. (2014). Neospora caninum infection during early pregnancy in cattle: how the isolate influences infection dynamics, clinical outcome and peripheral and local immune responses. Vet Res. 30;45:10.
  • He, X., Gong, P., Wei, Z., Liu, W., Wang, W., Li, J., et al. (2017). Peroxisome proliferator-activated receptor-γ-mediated polarization of macrophages in Neospora caninum Exp.Parasitol. 178, 37-44.
  • García-Sánchez, M., Jiménez-Pelayo, L., Horcajo, P., Regidor-Cerrillo, J., Ólafsson, E. B., Bhandage, A. K., et al. (2019). Differential responses of bovine monocyte-derived macrophages to infection by Neospora caninum isolates of high and low virulence. Front. Immunol. 10, 915.

Section 4, “Ultrastructure of caninum and functions of essential organelles”:

  • This section should be revised by the authors because several proteins are not cited in the text, such as NcMIC19 and NcMIC26 (Reid et al., 2012); NcMIC2 (Lovett et al., 2000); NcTRAP2; NcRON2, NcRON4, NcRON5 and NcRON8, etc. Moreover, there is no information about the initial host cell recognition of the parasite that is mediated by parasite surface antigens, such as NcSAG1 and NcSRS2 proteins (Hemphill & Gottstein, 1996).
  • L101: Delete merozoite.
  • L11: change AMA1” by NcAMA1”
  • L124: Change “Numerous ROP proteins of caninum are identified as virulence factors” by “Numerous ROP proteins of N. caninum are identified as potential virulence factors”.
  • L125-126: Not all proteins cited are pseudokinases.

Section 7.1., “Recognition receptors”:

  • L245-246: “So far 12 functional TLRs, subdivided based on cellular location, have been identified in mice (10 in human)”. What about TLRs in bovine? Please, add this information.
  • L263: Dectin-1 is a c-type lectin receptor (CLR). This should be described in the text.
  • Some general information about NFAT and L-arginine pathways should be added.

Section 7.3., “Cellular immune responses” section:

  • This part needs to be reorganized because the information given by the authors does not aid to understand the immune response during Neospora I recommend describing separately results by type of experiment (in vitro or in vivo) and by animal species.
  • L419: Move “Both CD4+ (helper) and CD8+ 419 (cytotoxic) T cells are determinant factors for the outcome of neosporosis [126,127]” to section 7.3.2.
  • L445-446 “Bovine macrophages infection with caninum (Spain7 and H1 isolates) associated with elevation of gene expression responsible for regulatory immune response and inflammatory processes”: It is unclear if the genes are from the parasite or from the host.

Section 7.3.2.2.1., “Effect of cytokine and oxidative compounds” section:

  • This part needs to be reorganized because is a list of unconnected data without a particular understanding.
  • L472: Please, include ROS data.
  • L518-522: Delete “In a number of studies using different immune cells from various animal models, caninum infection showed high efficiency in triggering extracellular trap like; the specific networks limiting the parasite proliferation and mediating the killing of parasite. This effect was noticed in case of bovine neutrophils [118], caprine monocytes [121], and canine neutrophils [122].” because this information is repetitive and redundant (L395 and Table 1).

Section 7.3.2.2.2. “Non-cytokine host peptide effectors” section:

  • It would be interesting to include here some information about cathelicidin peptides: “Characteristic pro-inflammatory cytokines and host defence cathelicidin peptide produced by human monocyte-derived macrophages infected with Neospora caninum (Boucher et al., 2018).

The manuscript needs language editing, just to point out some as examples:

  • L17: change “numerous key factors from both parasite and host sides are producing and interacting for…” by “numerous key factors from both parasite and host sides are produced and interact for…”
  • L30-31: change “The disease affecting large scale of warm-blooded animals and is transmitted by oral rout…” by “The disease affects large scale of warm-blooded animals and is transmitted by oral route…”
  • L36: change “However, the outcomes of Neospora infection are greatly variables ….” by “…the outcomes of Neospora infection are greatly variable…”
  • L42-43: Change “… tachyzoites differentiated quickly to bradyzoites” by “… “tachyzoites differentiate quickly to bradyzoites”.
  • L81, L83, L147: check the font size
  • L78 and Table 3 (reference 182): Change Neospora caninum by caninum
  • L276: Change “In this regard, Spain 7 and Spain 1H could activate …” by “In this regard, Nc-Spain7 and Nc-Spain1H isolates”.
  • L278: Change “bone marrow derived macrophages” by “bone marrow derived macrophages (BMDM)”
  • L328: Add a dot at the end of the sentence: “The canonical pathway is stimulated by a wide range of stimuli, such as TNF-α and IL-1β [101].”
  • L445: Change “Spain7 and H1 isolates” by “Nc-Spain7 and Nc-Spain1H isolates 1H.
  • L451-452: Add + to CD4 y CD8: CD4+, CD8+.
  • L485: “fat-tailed dunnart” should not be in italics or even this information should be delete because fat-tailed dunnart is not a relevant animal model for canium.
  • L506-508: Change “but also anti-inflammatory ones may play an essential role in regulating immunity against caninum infection particularly IL-10 and regulatory IL-4” by “…but also anti-inflammatory ones may play an essential role in regulating immunity against N. caninum infection particularly IL-4 and regulatory IL-10”.
  • L538: There are two dots at the end of the sentence.
  • L567-568: please, restructure “Summary of specific antibody production against caninum and specific molecules is summarized in table 4”.
  • Please, change “strain” by “isolate” throughout the text and tables.
  • Parasite isolate terminology should be homogenized throughout the text and tables.
  • Please, check and homogenize terminology: WT or wt?; IFN-γ or IFNγ?; CD4+ or CD4+ T cells?.
  • Please, change all the italics for Neospora y caninum throughout the text and tables.
  • Please, check all the abbreviations throughout the text and tables.

Table 1:

  • The data presentation should be revised, improved and reorganized.
  • The terms “Neospora molecule” and “ caninum infected animals under field conditions” should be replaced by others more appropriate.
  • Parasite terminology for the isolates should be homogenized. For example: caninum tachyzoites (Nc-1 isolate)/ N. caninum tachyzoites (Nc-1)/ Tachyzoites from N. caninum (Nc-1)/ N. caninum (Nc-1 isolate) tachyzoites/ N. caninum tachyzoites of the NC-1 isolate/ Tachyzoites of N. caninum-1 strain/ Tachyzoites of the Nc-1 strain, and etc. The same for parasite proteins (N. caninum GRA6/ NcGRA7/ N. caninum Nc-1 14-3-3 protein, etc.) and receptors (CCR5/ CCR5 receptor; NLR 3/ NLR3; TLR2 and TLR4/ TLRs 2 and 9/ TLR4- and IL-12Rβ2/ TLR3, 7 and 8, 9, etc.)
  • Reference 81: Please, indicate the role of TLR2 and TLR4 in the “Impacts and outcomes” column. The sentence does not say anything.
  • Reference 83: TLR-/- or TLR2-/- ?
  • Please, explain PMN, PM, ESAs, BMDDC, OML. These should be reflected in a footnote to the table.
  • Reference 95: Please, pephrase and correct the sentence: “In the in caninum-infected CCR5-/- mice, Increased mortality and neurological dysfunctions were observed, poor migration of dendritic cells and NKT cells to the site of infection”.
  • Reference 97: change “Dectin-1 presented a reduction in the parasite load…” by “Dectin-1-/- mice presented a reduction in the parasite load…”
  • Reference114: Rephrase and correct the sentence: “Phosphorylated NF-kB/p65 was observed in peritoneal macrophages treated with rNc14-3-3, and the protein level of NF-kB/p65”.
  • Reference 116: change CD+4 and CD+8 by CD4+ y CD8+
  • References 118, 120, 121 and 122 describe the same mechanisms and should be merged in the same row.
  • Reference 124: change “reduced the inhibitory effects induced by IFNIFN-γ” by “reduced the inhibitory effects induced by IFN-γ”.

Table 2:

  • The data presentation should be revised, improved and reorganized.
  • The terms “Type of host”, “Neospora molecule” and “ caninum infected animals under field conditions” should be replaced by others more appropriate.
  • Check terminology for parasite isolates, proteins, immune cells (macrophage/ macrophages), etc.
  • Reference 126: Change “CD+4 and CD+8” by CD4+ and CD8+.
  • Reference 134: CD14 and CD21are not markers for T cells.

Table 3:

  • The data presentation should be revised, improved and reorganized.
  • The terms “Type of host”, “Neospora molecule” and “ caninum infected animals under field conditions” should be replaced by others more appropriate.
  • Please, check terminology for parasite isolates.
  • Reference 148: change “ caninum – Infection” by “N. caninum – infection”
  • Please, homogenize terminology: weeks pi or wpi?
  • Reference 170, 184, 185: “Neospora tachyzoite”. What isolate?
  • Reference 176. Change “…IFN-g secretion” by IFN-γ secretion”.
  • Reference 121. Please, define ETs.
  • Data corresponding to references 186, 187 and 188 should be deleted.

Author Response

General author response

Comments of the reviewers were highly insightful and enabled us to greatly improve the quality of our manuscript. In the following pages are our point-by-point responses to each of the comments of the reviewers. Revisions in the text of the manuscript and response in revision note are shown with blue fonts. In addition, we submitted another file with tracking changes to check all revision made in the manuscript. Because of several additions and deletions, line and page numbers specified by reviewers have been changed. Additionally, the length of some parts was altered and some new references were added. We hope that the revisions in the manuscript and our accompanying responses will be sufficient to make our manuscript suitable for publication in the journal of “pathogens”.

The topic of the manuscript is relevant and original because is the first review about molecular mechanisms and host signaling pathways related to immune responses in Neospora caninum infection. Authors have conducted a very deep and complete review, providing interesting data. Therefore, this manuscript could be of interest to the readership of Pathogens.

Answer: We strongly appreciate reviewer 1 comments and encouraging words about our review.

However, some concerns should be addressed to improve the quality of the manuscript:

The first sections of the manuscript about general comments on bovine neosporosis are extremely long. I recommend reducing in length the following sections: “Historical view and evolution of Neospora caninum”; “Host range, transmission, and life cycle” and “Pathogenesis and clinical neosporosis in different animals”.

Answer: We already deleted some sentences from these sections without altering the overall meaning.

  1. Historical view and evolution of N. caninum
  2. Host range, transmission, and life cycle.
  3. Pathogenesis and clinical neosporosis in different animals.

No references for the “Overview of recent researches on machinery of neosporosis” section have been included.

Answer: We added some references for overview section. (page 2, lines 70, 74)

L33: delete “stormy”.

Answer: We deleted this word.

L36-37: Please, include “variables depending on the parasite”.

Answer: We included. (page 1, line 36)

Several studies that have provided important results about the topic of the present review have not been cited by the authors. I would encourage the authors to consider the following studies:

Answer: We added all the following studies in our study in text and tables.

Strohbusch, M., Muller, N., Hemphill, A., Margos, M., Grandgirard, D., Leib, S., et al. (2009). Neospora caninum and bone marrow-derived dendritic cells: parasite survival, proliferation, and induction of cytokine expression. Parasite Immunol. 31, 366-72.

Answer: Done (page 15, line 492, reference 145)

Feng, X., Zhang, N., and Tuo, W. (2010). Neospora caninum tachyzoite- and antigen-stimulated cytokine production by bone marrow-derived dendritic cells and spleen cells of naive BALB/c mice. J.Parasitol. 96, 717-23

Answer: Done (page 15, line 492, reference 146)

Dion, S., Germon, S., Guiton, R., Ducournau, C., and Dimier-Poisson, I. (2011). Functional activation of T cells by dendritic cells and macrophages exposed to the intracellular parasite Neospora caninum. Int.J.Parasitol. 41, 685-95.

Answer: Done (page 16, line 494, reference 147)

Regidor-Cerrillo J, Arranz-Solís D, Benavides J, Gómez-Bautista M, Castro-Hermida JA, Mezo M, Pérez V, Ortega-Mora LM, González-Warleta M. (2014). Neospora caninum infection during early pregnancy in cattle: how the isolate influences infection dynamics, clinical outcome and peripheral and local immune responses. Vet Res. 30;45:10.

Answer: Done (page 16, line 514, reference 158)

He, X., Gong, P., Wei, Z., Liu, W., Wang, W., Li, J., et al. (2017). Peroxisome proliferator-activated receptor-γ-mediated polarization of macrophages in Neospora caninum Exp.Parasitol. 178, 37-44.

Answer: Done (page 7, line 323, reference 106)

García-Sánchez, M., Jiménez-Pelayo, L., Horcajo, P., Regidor-Cerrillo, J., Ólafsson, E. B., Bhandage, A. K., et al. (2019). Differential responses of bovine monocyte-derived macrophages to infection by Neospora caninum isolates of high and low virulence. Front. Immunol. 10, 915.

Answer: Done (page 15, line 485, reference 144)

Section 4, “Ultrastructure of caninum and functions of essential organelles”:

This section should be revised by the authors because several proteins are not cited in the text, such as NcMIC19 and NcMIC26 (Reid et al., 2012); NcMIC2 (Lovett et al., 2000); NcTRAP2; NcRON2, NcRON4, NcRON5 and NcRON8, etc. Moreover, there is no information about the initial host cell recognition of the parasite that is mediated by parasite surface antigens, such as NcSAG1 and NcSRS2 proteins (Hemphill & Gottstein, 1996).

Answer: We added some of these proteins and relevant studies in our study especially those well characterized and studied:

For microneme proteins as follows:

“……InN. caninum, NcMIC2 [30], NcMIC3 [20], NcMIC4 [31], NcMIC6 [32], NcMIC19 and NcMIC26 [33] can initiate motility, attachment and invasion of the host cells…….” (page 3, lines 118-120).

For rhoptry proteins as follows:

“Numerous ROP proteins of N. caninumare identified as potential virulence factors, with some considered active protein kinases as NcROP18 [38], whilst others are deficient in the known kinase-like catalytic domain and are regarded as pseudokinases such as, NcROP2Fam-1, NcROP5, NcROP40 [39], and NcROP16 [40]. Numerous RON proteins of N. caninumhad been already identified using immunoproteomic analysis such as NcRON4, NcRON5 [41] and NcROP1, 8, 30, and NcRON2, 3, 4, 8 [42].” (page 3, lines 131-136).

For surface proteins as follows:

“Two major surface antigens were identified and widely studied; N. caninum surface antigen 1 (SAG1, previously named NcP36), and SAG-related sequence protein 2 (SRS2, previously named NcP43). NcSAG1 is anchored on the surface of tachyzoites by glycosylphosphatidylinositol (GPI) anchor [25], while NcSAS2 has also GPI anchor [26], and is expressed in both tachyzoites and bradyzoites [24].” (page 3, lines 110-114).

L101: Delete merozoite.

Answer: Done

L11: change AMA1” by NcAMA1”

Answer: Done

L124: Change “Numerous ROP proteins of caninum are identified as virulence factors” by “Numerous ROP proteins of N. caninum are identified as potential virulence factors”.

Answer: Done (page 3, line 131)

L125-126: Not all proteins cited are pseudokinases.

Answer: We revised this part as follows:

“Numerous ROP proteins of N. caninumare identified as potential virulence factors, with some considered active protein kinases as NcROP18 [38], whilst others are deficient in the known kinase-like catalytic domain and are regarded as pseudokinases such as, NcROP2Fam-1, NcROP5, NcROP40 [39], and NcROP16 [40].” (page 3, lines 131-134)

Section 7.1., “Recognition receptors”:

L245-246: “So far 12 functional TLRs, subdivided based on cellular location, have been identified in mice (10 in human)”. What about TLRs in bovine? Please, add this information.

Answer: We added this information as follows:

“In cattle, the majority of work on bovine toll-like receptors (TLRs) concentrates on TLRs 2 and 4. However, homologues of human TLRs 1–10 were exist within cattle homologues sharing at least 95% nucleotide sequence identity [81].” (page 6, lines 252-255)

L263: Dectin-1 is a c-type lectin receptor (CLR). This should be described in the text.

Answer: We added this information. (page 7, lines 272-273).

Some general information about NFAT and L-arginine pathways should be added.

Answer: We added this information.

“Nuclear factor of activated T cell (NFAT) proteins were first identified in T-cells as transcriptional activators of IL-2, a key regulator of T cell immune response [112]. There are four types of proteins in the NFAT gene family: NFATc1, NFATc2, NFATc3, and NFATc4. Such proteins are mediated by the phosphatase calcineurin that dephosphorylates NFAT proteins to expose their nuclear localization signals, thus inducing the nuclear translocation of NFAT proteins. Subsequently, in the nucleus, NFAT proteins interact with other factors to trigger the target gene expression, essential for many biological functions [113]” (page 8, lines 347-353)

“The L-arginine-nitric oxide pathway is a regulating mechanism triggered by the formation of nitric oxide (NO) from the amino acid L-arginine by the nitric oxide synthase. NO is produced abundantly in tissues of cardiovascular and nervous systems. In all these tissues the L-arginine: NO pathway acts as a transduction mechanism for the soluble guanylate cyclase regulated by NO. In addition, NO can be secreted from numerous immune cells such as macrophages, neutrophils, and other cells suggesting its role in the host defense mechanism either against tumor cells or invasive organisms [124,125].” (page 9, lines 392-398)

Section 7.3., “Cellular immune responses” section:

This part needs to be reorganized because the information given by the authors does not aid to understand the immune response during Neospora I recommend describing separately results by type of experiment (in vitro or in vivo) and by animal species.

Answer: This part is reorganized and substantially revised to become more simple, understandable and clear. Separation of results in data presentation in text or in tables according to type of experiment (in vitro or in vivo) is very difficult for these reasons. First, many studies had been investigated the role of cells or effector molecules using both in vitro and in vivo assays. Second, in the part of data presentation of table in the text, we considered the presentation according to type of cells or molecule, animal species, then by reference, so it is difficult to add additional factors for presentation or discussion. However, we considered the more elucidation based o type of experiments or assay in a way that enabled avoiding repetitive and complicated writing style.

L419: Move “Both CD4+ (helper) and CD8+ 419 (cytotoxic) T cells are determinant factors for the outcome of neosporosis [126,127]” to section 7.3.2.

Answer: We moved this sentence. (page 16, lines 498-499).

L445-446 “Bovine macrophages infection with caninum (Spain7 and H1 isolates) associated with elevation of gene expression responsible for regulatory immune response and inflammatory processes”: It is unclear if the genes are from the parasite or from the host.

Answer: We clarified this information as follows:

“In vitroinfection of bovine macrophages with N. caninum(Nc-Spain7 and Nc-SpainH1 isolates) associated with elevation of the parasite gene expression responsible for regulatory immune response and inflammatory processes from the host [143].”  (page 15, lines 479-482).

Section 7.3.2.2.1., “Effect of cytokine and oxidative compounds” section:

This part needs to be reorganized because is a list of unconnected data without a particular understanding.

Answer: This part is reorganized, revised substantially and some sentences have been added to connect information as follows:

“Cytokines, ROS and reactive nitrogen species (RNS) are considered to be important factors in the pathogenesis of N. caninuminfections. These compounds are secreted by the host as response to control the proliferation and dissemination of the parasite.The type and degree of response depends on several factors related to the host or the parasite factors or even on experimental approach.”(page 20, lines 525-528).

“These results indicated the secretion of IFN-γ in various cells of different hosts in a response against N. caninuminfection regardless of used isolates. Thus, this cytokine is encountered as a vital molecule for the host resistance against N. caninuminfection.” (page 20, lines 551-554)

“Thus, proinflammatory cytokines and oxidative mediators are critical for controlling the neosporosis via its direct destructive effects on the parasite.” (page 20, lines 565-567).

‘Several studies have found that bioactive molecules can mediate the host defense against N. caninuminfection, and in some cases ameliorate the fatal consequences of neosporosis. These biomolecules are gaining wide attention for their anti-N. caninumactivities.” (page 21, lines 588-590).

L472: Please, include ROS data.

Answer: We added information on ROS as follows:

“Also, in a similar study investigated the same host cell and parasite isolate, elevated levels of reactive oxygen species (ROS) and IL-12 was recorded in Nc-SpainH1 than Nc-Spain7 or non-infected cells. In addition, infected macrophages with both isolates showed lower expression of MHC Class II, CD86 and CD1b molecules than non-infected cells [144].” (page 15, lines 482-485).

L518-522: Delete “In a number of studies using different immune cells from various animal models, caninum infection showed high efficiency in triggering extracellular trap like; the specific networks limiting the parasite proliferation and mediating the killing of parasite. This effect was noticed in case of bovine neutrophils [118], caprine monocytes [121], and canine neutrophils [122].” because this information is repetitive and redundant (L395 and Table 1).

Answer: We apologize to reviewer 1 for this confusing point. Actually, the sentences in both parts are involving same references but the data is totally different. The sentence in L395 “Even in ruminant animal model, in cattle and goat, activation of macrophages or neutrophils p38 MAPK and ERK1/2- with N. caninumlive tachyzoites triggered the formation of extracellular trap-like structures (ETs)[131-133].” (page 9, lines 422-424). Here, we focused on MAPK/ERK signaling pathways regulating this effect in different animal cells and the data is presented in table 1.

While sentence in page 21 , lines 581-584 “In a number of studies using different immune cells from various animal models, N. caninuminfection showed high efficiency in triggering extracellular trap like; the specific networks limiting the parasite proliferation and mediating the killing of parasite. This effect was noticed in case of bovine neutrophils [131], caprine monocytes [133], and canine neutrophils [134]” as shown above is discussing the interaction of parasite with neutrophils of different animals but not at molecular interface because these data are already discussed separately in cited papers. Because data presentation involves different points and their existence is important in table 1 and table 2, it is very difficult to delete and we kindly ask reviewer 1 to accept these information that further clarify and confirm efficient manipulation of N. caninumfor different components of host immunity particularly after revision of sentences and information. 

Section 7.3.2.2.2. “Non-cytokine host peptide effectors” section:

It would be interesting to include here some information about cathelicidin peptides: “Characteristic pro-inflammatory cytokines and host defence cathelicidin peptide produced by human monocyte-derived macrophages infected with Neospora caninum (Boucher et al., 2018).

Answer: We added this information as follows:

“Also, a previous report indicated that cathelicidins (host defence peptide in human cells) showed higher expression levels in macrophages infected with live tachyzoites of N. caninumthan naïve cells. This elevation was associated with increment in TNF-α, IL-1β, IL-8, IFN-γ cytokine and reduced parasite internalization in naïve macrophages [207].” (page 21, lines 603-606)

The manuscript needs language editing, just to point out some as examples:

Answer: Grammatical and spelling errors have been corrected throughout the MS.

L17: change “numerous key factors from both parasite and host sides are producing and interacting for…” by “numerous key factors from both parasite and host sides are produced and interact for…”

Answer: Done (page 1, line 18)

L30-31: change “The disease affecting large scale of warm-blooded animals and is transmitted by oral rout…” by “The disease affects large scale of warm-blooded animals and is transmitted by oral route…”

Answer: Done (page 1, line 30)

L36: change “However, the outcomes of Neospora infection are greatly variables ….” by “…the outcomes of Neospora infection are greatly variable…”

Answer: Done (page 1, line 36)

L42-43: Change “… tachyzoites differentiated quickly to bradyzoites” by “… “tachyzoites differentiate quickly to bradyzoites”.

Answer: Done (page 2, line 42)

L81, L83, L147: check the font size.

Answer: Done (page 2, line 81)

N.B. although we adjusted font size and changed scientific names (T. gondiiand N. caninum) of part in page 2, lines 80-83 several times with different ways and saved changes, but sometimes they changed and showed differences than other text again for unknown reason (may be related by journal format editing).

L78 and Table 3 (reference 182): Change Neospora caninum by caninum.

Answer: Done(Neospora caninumchanged toN. caninum)

L276: Change “In this regard, Spain 7 and Spain 1H could activate …” by “In this regard, Nc-Spain7 and Nc-Spain1H isolates”.

Answer: We revised throughout the MS.

L278: Change “bone marrow derived macrophages” by “bone marrow derived macrophages (BMDM)”

Answer: We revised throughout the MS.

L328: Add a dot at the end of the sentence: “The canonical pathway is stimulated by a wide range of stimuli, such as TNF-α and IL-1β [101].”

Answer: Done(page 8, line 343)

L445: Change “Spain7 and H1 isolates” by “Nc-Spain7 and Nc-Spain1H isolates 1H.

Answer: We revised throughout the MS.

L451-452: Add + to CD4 y CD8: CD4+, CD8+.

Answer: We revised throughout the MS.

L485: “fat-tailed dunnart” should not be in italics or even this information should be delete because fat-tailed dunnart is not a relevant animal model for canium.

Answer: We changed to italics, we kept this information, because we want to illustrate the N. caninum-host interaction even in some susceptible hosts as already discussed in some parts for human cell. (page 20 , line 541)

L506-508: Change “… but also anti-inflammatory ones may play an essential role in regulating immunity against caninum infection particularly IL-10 and regulatory IL-4” by “…but also anti-inflammatory ones may play an essential role in regulating immunity against N. caninum infection particularly IL-4 and regulatory IL-10”.

Answer: Done (page 20, line 569)

L538: There are two dots at the end of the sentence.

Answer: Deleted

L567-568: please, restructure “Summary of specific antibody production against caninum and specific molecules is summarized in table 4”.

Answer: Done (page 29, lines 636-637)

From “Summary of specific antibody production againstN. caninumand specific molecules is summarized in table 4.”

To

“Summary of specific antibody production (total IgG and subtypes or IgM) against N. caninumand its specific molecules is summarized in table 4.”

Please, change “strain” by “isolate” throughout the text and tables.

Answer: We have changed according to reviewer 1 request throughout the MS in text and tables.

Parasite isolate terminology should be homogenized throughout the text and tables.

Answer: We have changed according to reviewer 1 request throughout the MS in text and tables.

Please, check and homogenize terminology: WT or wt?; IFN-γ or IFNγ?; CD4+ or CD4+ T cells?.

Answer: Done also for other terminologies as dpi, wpi, TNF-α throughout the MS in text and tables.

We used WT, IFN-γ, CD4+T cells, dpi, wpi, TNF-α. Other examples can be found throughout MS.

Please, change all the italics for Neospora y caninum throughout the text and tables.

Answer: Done

Please, check all the abbreviations throughout the text and tables.

Answer: We confirmed all abbreviation in throughout the MS by full description at first appearance and also added a separate section for abbreviations at the end of MS.

Table 1:

The data presentation should be revised, improved and reorganized.

Answer: Substantial changes have been done in table 1. Changes are indicated in blue colored fonts and can be checked also in file with tacking changes.

The terms “Neospora molecule” and “ caninum infected animals under field conditions” should be replaced by others more appropriate.

Answer: in heading of table 1, terms of “type of host” changed to “host species”, and “Neospora molecule” to “parasite or its molecule”.

And “ caninum infected animals under field conditions” replaced by “N. caninum innaturally infected animals”

Parasite terminology for the isolates should be homogenized. For example: caninum tachyzoites (Nc-1 isolate)/ N. caninum tachyzoites (Nc-1)/ Tachyzoites from N. caninum (Nc-1)/ N. caninum (Nc-1 isolate) tachyzoites/ N. caninum tachyzoites of the NC-1 isolate/ Tachyzoites of N. caninum-1 strain/ Tachyzoites of the Nc-1 strain, and etc. The same for parasite proteins (N. caninum GRA6/ NcGRA7/ N. caninum Nc-1 14-3-3 protein, etc.) and receptors (CCR5/ CCR5 receptor; NLR 3/ NLR3; TLR2 and TLR4/ TLRs 2 and 9/ TLR4- and IL-12Rβ2/ TLR3, 7 and 8, 9, etc.).

Answer: Done for above-mentioned and all other parasite terminology are revised and homogenized

We usedN. caninum(Nc-1)/ N. caninumtachyzoites (Nc-1). NcGRA7, Nc14-3-3, CCR5, NLR3, TLR2, IL-12Rβ2/ TLRs 3, 7 and 8, 9, etc.). Other examples can be found throughout MS.

Reference 81: Please, indicate the role of TLR2 and TLR4 in the “Impacts and outcomes” column. The sentence does not say anything.

Answer: we added as follows: NcGPI induced stimulation of TLR2 and TLR4 from HEK cells,…..” (table 1, reference 89)

Reference 83: TLR-/- or TLR2-/- ?.

Answer: Revisedas TLR2-/-(table 1, reference 91) and throughout MS.

Please, explain PMN, PM, ESAs, BMDDC, OML. These should be reflected in a footnote to the table. Because tables are full with abbreviations,

Answer: We added in footnote of table 1 and we prepared a separate part for abbreviations at the end of MS (except for PM where we used full name peritoneal macrophages without abbreviation in tables and text). In addition we prepared a separate section for abbreviation at the end of MS.

Reference 95: Please, pephrase and correct the sentence: “In the in caninum-infected CCR5-/- mice, Increased mortality and neurological dysfunctions were observed, poor migration of dendritic cells and NKT cells to the site of infection”.

Answer: Revised and corrected (reference 103)

“In the N. caninum-infected CCR5-/-mice, increased mortality and neurological dysfunctions, poor migration of DCs and NKT cells to the site of infection were observed.”

Reference 97: change “Dectin-1 presented a reduction in the parasite load…” by “Dectin-1-/- mice presented a reduction in the parasite load…”

Answer: Done (table 1, reference 105)

Reference114: Rephrase and correct the sentence: “Phosphorylated NF-kB/p65 was observed in peritoneal macrophages treated with rNc14-3-3, and the protein level of NF-kB/p65”.

Answer: We revised as follows:

“Phosphorylated NF-kB/p65 was observed in peritoneal macrophages treated with rNc14-3-3.” (table 1, reference 127)

Reference 116: change CD+4 and CD+8 by CD4+ y CD8+

Answer: We changed to CD4+and CD8+(table reference 129) and throughout the MS.

References 118, 120, 121 and 122 describe the same mechanisms and should be merged in the same row.

Reference 124: change “reduced the inhibitory effects induced by IFNIFN-γ” by “reduced the inhibitory effects induced by IFN-γ.

Answer: we deleted additional IFN (table 1, reference 137)

Table 2:

The data presentation should be revised, improved and reorganized.

Answer: Substantial changes have been done in table 2. Changes are indicated in blue colored fonts and can be checked also in file with tacking changes.

The terms “Type of host”, “Neospora molecule” and “ caninum infected animals under field conditions” should be replaced by others more appropriate.

Answer: in heading of table 2, terms of “type of host” changed to “host species”, and “Neospora molecule” to “parasite or its molecule”

And “ caninum infected animals under field conditions” replaced by “N. caninum innaturally infected animals”

Check terminology for parasite isolates, proteins, immune cells (macrophage/ macrophages), etc. Answer: Done for above-mentioned terms and others throughout the tables and text.

For cells we used macrophages, monocytes, dendritic cells, T cells, CD4+and CD8+T cells, natural killer T cells.

For Neospora caninumisolate: Nc-1, Nc-Liv, Nc-Spain1H, ….

For proteins: NcGRA6, NcCyp, ….

Reference 126: Change “CD+4 and CD+8” by CD4+ and CD8+.

Answer: We revised to CD4+and CD8+T cells (table 2, reference 149)

Reference 134: CD14 and CD21are not markers for T cells.

Answer: we deleted CD14 and CD21 and kept only CD4 and CD8 as markers to T cells (table 2, reference 150)

N.B. We added in footnote of table 2 and we prepared a separate part for abbreviations at the end of MS.

Table 3:

The data presentation should be revised, improved and reorganized.

Answer: Substantial changes have been done in table 3. Changes are indicated in blue colored fonts and can be checked also in file with tacking changes.

The terms “Type of host”, “Neospora molecule” and “ caninum infected animals under field conditions” should be replaced by others more appropriate.

Answer: in heading of table 3, terms of “type of host” changed to “host species”, and “Neospora molecule” to “parasite or its molecule”

And “ caninum infected animals under field conditions” replaced by “N. caninum innaturally infected animals”

Please, check terminology for parasite isolates.

Answer: Terms for parasite isolates and others are revised and homogenized.

For parasite isolate, we homogenized as follows: Nc-1, Nc-Liv, Nc-Nowra, Nc-Spain7, Nc-Spain1H, Nc-SweB1, Nc-Bahia in all tables and throughout the MS.

Reference 148: change “ caninum – Infection” by “N. caninum – infection”

Answer: Done (table 3, reference 166)

Please, homogenize terminology: weeks pi or wpi?

Answer: We have used wpi according to reviewer 1 request throughout the MS in text and tables.

Reference 170, 184, 185: “Neospora tachyzoite”. What isolate?

Answer: Type of isolate was added in all data of all tables in column of parasite or its molecule.

Reference 176. Change “…IFN-g secretion” by IFN-γ secretion.

Answer: Done (reference 194)

Reference 121. Please, define ETs.

Answer: It is already defined as “extracellular trap in first column”, in addition we added in footnote for table 3 and we prepared a separate part for abbreviations at the end of manuscript.

Answer: Data corresponding to references 186, 187 and 188 should be deleted.

Because the purpose of this study is the explanation of molecular mechanism of N. caninuminfection and hosts or parasite factors, and information from these references showed definite interaction with N. caninumwith some molecules of non-immune nature. Reference numbers have been changed to 204, 205 and 206, respectively.

Reviewer 2 Report

This is a well-organized and written article. 

Author Response

We would like to thank reviewer 2 for acceptance of our manuscript.

Reviewer 3 Report

The manuscript ID: pathogens-776393, titled “From signaling pathways to distinct immune responses: Key factors for establishing or combating Neospora caninum infection in different susceptible hosts” is well-written, clear, consistent and easy to understand. Definitely, it deserves to be published.  I briefly suggest some modifications and basic recommendations.

  1. Introduction: It is important to highlight that Neospora caninum is not a zoonotic apicomplexan parasite, as others (e.g. Toxoplasma gondii). This aspect is also not explicitly mentioned in Section 3 (“Historical view and evolution of Neospora caninum”). This is one of the first questions from the non-topic readers.
  2. Section 5. “Host range, transmission, and life cycle”:
    1. Lines 147-149; Intermediate hosts: Apart from cervids, birds, and foxes, authors should mention other wildlife species that are truly involved in the parasite life-cycle. I explicitly mean to wildlife species that really experience clinical neosporosis, play an important role in the transmission of N. caninum and consequently, are relevant to the epidemiology [e. g. buffaloes, wolves (Canis lupus), Australian Dingo (C. lupus dingo) or Coyote (C. latrans)].
  3. Section 7. “Immunity”: Authors initially described the recognition receptors involved in both innate and adaptive immunity, but what about the relevance of the major immunodominant surface antigens (SAG) of N. caninum tachyzoites as the initial low affinity host-parasite contact spot? They play a key role in the modulation of host immunity and parasite virulence when they first bind to the host cell surface glycosaminoglycan’s (GAGs). A detailed explanation of the early effector immune mechanisms is vital to understand the signaling pathways well-described posteriorly.
  4. Are there reports on immunity and signaling pathways performed on other key species apart of cattle, goat, sheep, dog and, mice (for example wildlife animal models)? If yes, please report some of them.
  5. The authors should briefly mention the importance of understanding the different cascades, signaling pathways and host-parasite effector mechanisms on the search of therapeutic drugs and vaccines against neosporosis. There are a great number of scientists worldwide, working to achieve this objective, with abundant literature available, who deserve to be cited.

Minor considerations:

  1. Figure 1 is very simple and easy to understand, however, not all the steps are clear and some points need to be improved: A better resolution of the picture is required. Moreover, the words “Macrophage, Dendritic cell” are overlapped with the cell shape. The term “Eradication” is not appropriate for the description of the destruction of the infected cells (please use another suitable word). An extra short legend inside the picture indicating the parasite stage (N. caninum tachyzoite) is mandatory (reinforces the description performed in the legend).
  2. Legend of figure 1 needs to be corrected, as some aspects are unclear, especially to facilitate the easy comprehension of non-topic readers:
    1. Please use the terms “N. caninum” or “Neospora spp.”, instead of only “Neospora” (for example in lines 593 and 596).
    2. TLR, NOD-like receptors and IFN-γ: It is necessary to point out their position in the image. If the authors reject to use so many words inside the text, a simple letter (a, b, c) inside the image might be useful enough.
    3. Activation of cellular NF-kB signaling pathway: It needs to be pointed in the image as well.
    4. Line 589: Which tachyzoite-antigenic proteins and molecules are uptaken by macrophages and dendritic cells? Please specify.

Author Response

General author response

Comments of the reviewers were highly insightful and enabled us to greatly improve the quality of our manuscript. In the following pages are our point-by-point responses to each of the comments of the reviewers. Revisions in the text of the manuscript and response in revision note are shown with blue fonts. In addition, we submitted another file with tracking changes to check all revision made in the manuscript. Because of several additions and deletions, line and page numbers specified by reviewers have been changed. Additionally, the length of some parts was altered and some new references were added. We hope that the revisions in the manuscript and our accompanying responses will be sufficient to make our manuscript suitable for publication in the journal of “pathogens”.

The manuscript ID: pathogens-776393, titled “From signaling pathways to distinct immune responses: Key factors for establishing or combating Neospora caninum infection in different susceptible hosts” is well-written, clear, consistent and easy to understand. Definitely, it deserves to be published. I briefly suggest some modifications and basic recommendations.

Answer: We highly appreciate the comments and encouraging words of reviewer 3

Introduction: It is important to highlight that Neospora caninum is not a zoonotic apicomplexan parasite, as others (e.g. Toxoplasma gondii). This aspect is also not explicitly mentioned in Section 3 (“Historical view and evolution of Neospora caninum”). This is one of the first questions from the non-topic readers.

Answer: We added this information as follows:

“Regarding human, although the infection in two rhesus monkeys (Macaca mulata) and specific antibodies to N. caninumin humans had been recorded, there is no evidence that N. caninum infection is zoonotic or induced clinical forms [1].” (page 5, line 229-231)

In addition, we already added this information in previous version of MS in introduction section as follows:

“Although no evidence for neosporosis in human, the disease recently gained a significant interest because of massive economic losses associating abortion in cattle [1]”(page 2, line 44)

Section 5. “Host range, transmission, and life cycle”:

Lines 147-149; Intermediate hosts: Apart from cervids, birds, and foxes, authors should mention other wildlife species that are truly involved in the parasite life-cycle. I explicitly mean to wildlife species that really experience clinical neosporosis, play an important role in the transmission of N. caninum and consequently, are relevant to the epidemiology [e. g. buffaloes, wolves (Canis lupus), Australian Dingo (C. lupus dingo) or Coyote (C. latrans)].

Answer: We added this information as follows:

“Sexual reproduction occurred only in the definitive host in animals of Canidaesuch as the dog [53], coyote [54], grey wolf [55], and dingo [56]. Theseanimals can experience clinical neosporosis, play an important role in the transmission of N. caninum,and consequently, are relevant to the epidemiology of the disease.” (page 4, lines 151-155).

Section 7. “Immunity”: Authors initially described the recognition receptors involved in both innate and adaptive immunity, but what about the relevance of the major immunodominant surface antigens (SAG) of N. caninum tachyzoites as the initial low affinity host-parasite contact spot? They play a key role in the modulation of host immunity and parasite virulence when they first bind to the host cell surface glycosaminoglycan’s (GAGs). A detailed explanation of the early effector immune mechanisms is vital to understand the signaling pathways well-described posteriorly.

Are there reports on immunity and signaling pathways performed on other key species apart of cattle, goat, sheep, dog and, mice (for example wildlife animal models)? If yes, please report some of them.

Answer: We added this information for initial parasite-host interaction as follows:

“In addition, the initial host cell recognition of the parasite that is mediated by parasite surface antigens is a critical step for establishing successful infection [24].

4.1. Surface antigens

  1. caninum tachyzoites surface antigens regulate the process of adhesion and invasion of host cells. Additionally, they contribute to the interaction of the parasite with the immune system, and subsequently the pathogenesis of the disease. Two major surface antigens were identified and widely studied; N. caninumsurface antigen 1 (SAG1, previously named NcP36), and SAG-related sequence protein 2 (SRS2, previously named NcP43). NcSAG1 is anchored on the surface of tachyzoites by glycosylphosphatidylinositol (GPI) anchor [25], while NcSAS2 has also GPI anchor [26], and is expressed in both tachyzoites and bradyzoites [24].” (page 3, line 104-114)

Regarding reports on immunity and signaling pathways in wild animals or others, all the found reports have been already discussed in previous version of manuscript.

The authors should briefly mention the importance of understanding the different cascades, signaling pathways and host-parasite effector mechanisms on the search of therapeutic drugs and vaccines against neosporosis. There are a great number of scientists worldwide, working to achieve this objective, with abundant literature available, who deserve to be cited.

Answer: We added a brief description especially for vaccine only at part of signaling pathways and at concluding remarks as follows:

“Deep understanding of differential mechanisms of N. caninum-host cell interactions at the molecular interface will greatly assist in developing control strategies against neosporosis. Indeed, abundant previous reports on such research field provided valuable knowledge on N. caninuminteraction with recognition receptors or signaling pathways. This knowledge had been already exploited to develop potent vaccines against neosporosis as reported by several studies using mice [90,91,] or in cattle [100]” (pages 9-10, lines 436-441)

“Nevertheless, specific roles of these pathways and the microenvironment accompanied N. caninum infection in different hosts particularly in cattle, dog and also mice still unknown. Hence, it is critical to perform more research studies and pursue a more complete understanding of the cascade-dependent signals which lead to immune response, consequently will lead to the development of fully effective control strategies against N. caninumin susceptible hosts including potent vaccines or effective therapeutic agents.” (page 30, lines 651-656)

Minor considerations:

Figure 1 is very simple and easy to understand, however, not all the steps are clear and some points need to be improved: A better resolution of the picture is required. Moreover, the words “Macrophage, Dendritic cell” are overlapped with the cell shape.

Answer: We changed to avoid overlapping. (Figure 1)

The term “Eradication” is not appropriate for the description of the destruction of the infected cells (please use another suitable word)

Answer: It is changed to “destruction”. (Figure 1)

An extra short legend inside the picture indicating the parasite stage (N. caninum tachyzoite) is mandatory (reinforces the description performed in the legend).

Answer: Extra legend was added for more illustrations. (Figure 1)

Legend of figure 1 needs to be corrected, as some aspects are unclear, especially to facilitate the easy comprehension of non-topic readers:

Answer: We revised and corrected some words. (figure 1 legend)

Please use the terms “N. caninum” or “Neospora spp.”, instead of only “Neospora” (for example in lines 593 and 596).

Answer: We changed “Neospora” to “N. caninum.” (figure 1 legend)

TLR, NOD-like receptors and IFN-γ: It is necessary to point out their position in the image. If the authors reject to use so many words inside the text, a simple letter (a, b, c) inside the image might be useful enough.

Answer: We illustrated using some symbols (figure 1)

Activation of cellular NF-kB signaling pathway: It needs to be pointed in the image as well.

Answer: We have pointed out in the legend of figure 1.

Line 589: Which tachyzoite-antigenic proteins and molecules are uptaken by macrophages and dendritic cells? Please specify.

Answer: We have referred in figure 1 legend as follows:

“Possible pathways of interaction of Neospora caninumtachyzoite or derived molecules and host immune effectors. In case of infection with N. caninum tachyzoites or parasite molecules (e.g. NcGPI, NcGRA6, NcGRA7 or NcCyp), are uptaken by macrophages or dendritic cells as professional antigen-presenting cells.” (page 31, lines 658-661).

Round 2

Reviewer 1 Report

Article ID: Pathogens-776393

Title: From signaling pathways to distinct immune responses: Key factors for establishing or combating Neospora caninum infection in different susceptible hosts

The authors have solved all major concerns and the manuscript has also been improved so that the present study deserves to be published.